# Integrated Approach for the Study of Urban Expansion and River Floods Aimed at Hydrogeomorphic Risk Reduction

Andrea Mandarino [1,*], Francesco Faccini [1,2], Fabio Luino [2], Barbara Bono [2] and Laura Turconi [2]

1   Department of Earth, Environment and Life Sciences, University of Genova, 16132 Genova, Italy; faccini@unige.it
2   National Research Council, Research Institute for Geo-Hydrological Protection, 10135 Torino, Italy; fabio.luino@irpi.cnr.it (F.L.); barbara.bono@irpi.cnr.it (B.B.); laura.turconi@irpi.cnr.it (L.T.)
*   Correspondence: andrea.mandarino@unige.it

**Abstract:** Urbanization in flood-prone areas is a critical issue worldwide. The historical floods, the urban expansion in terms of building footprint, the extent and construction period of inundated buildings with reference to two representative floods (5–6 November 1994 and 24–25 November 2016), and the ground effects and dynamics of these events were investigated in the cities of Garessio, Ceva, and Clavesana, along the Tanaro River (NW Italy). An integrated approach based on historical data analysis, photograph interpretation, field surveys, and GIS investigations was adopted, and novel metrics for quantitative analysis of urbanization and flood exposure at the individual-building scale were introduced. The considered cities were hit by damaging floods several times over the last centuries and experienced an increase in built-up surface after the mid-19th century, especially between the 1930s and 1994. The 1994 and 2016 high-magnitude floods highlighted that urban expansion largely occurred in flood-prone areas, and anthropogenic structures conditioned flood propagation. One of the rare Italian cases of the relocation of elements exposed to floods is documented. This research aims to emphasize the relevance of information on past floods and urbanization processes for land planning and land management and the need for land use planning for flood control to forbid new urban expansion in potentially floodable areas. The outcomes represent an essential knowledge base to define effective and sustainable management measures to mitigate hydrogeomorphic risk.

**Keywords:** historic floods; flood hazard; flood exposure; geomorphic effectiveness; geomorphic response; levee effect; urbanization; relocation; Tanaro River

## 1. Introduction

Fluvial floods, namely, overflows of water outside the riverbed, occur frequently in large areas of the world and are one of the natural hazards claiming the highest number of casualties in Europe [1–6]. These processes occur due to prolonged rainfall, which can affect large areas, or due to brief and intense rainfall, which can affect restricted areas of a few square kilometers with extremely rapid hydrological responses [7,8]. According to the EM-DAT database, since the late 1950s, at least 455 flood disasters, including both riverine and flash floods, have been recorded in Europe, 34 of which were in Italy [9].

The frequency of major floods in many places around the world seems to be increasing [4,10–12]. Climate change has been recognized as a significant driver of this change [13–15]. However, more importantly, anthropogenic interventions along riverbeds and over catchments have impacted worldwide upon the severity and consequences of floods [16–18], often increasing river-related risks because of riverbed channelization and riverine-area occupation [19–24].

Floodplains have always been attractive places for urban development due to their flat morphology, fertile soil, favorable exposure to solar radiation, easy access, and the possibility of utilizing groundwater [25,26]. Similarly, in mountain areas, valley floors have

been considered preferential areas for urban settlement growth primarily for their plain surface availability and easy access. Until the industrial era, land use over floodplains and valley floors was fundamentally tied to agricultural activities, while significant urban expansion began from the 19th century onward, with a further decisive impulse from the mid-20th century, now largely recognized as the beginning of the Anthropocene [27,28].

This rapid, often ongoing, urbanization of floodplains and valley floors has caused population and capital to become increasingly exposed to flood events [29–32]. These are natural and common phenomena associated with river dynamics that become 'natural disasters' when they impact humans and their activities [33–35]. Planning urban development from the perspective of managing floods with the aim of ensuring the safety and wellbeing of people and preserving the natural environment is one of the main responsibilities of public authorities [36,37]. Flood hazard maps and flood risk maps showing the potential adverse consequences associated with different flood scenarios, along with flood risk management plans at the catchment scale, are essential tools for achieving these goals, as stated by the European regulation in force [36]. In this light, the investigation of flood-prone areas and both potential and known dynamics of riverine floods has increasingly assumed a very relevant importance worldwide, such as the analysis of exposure and vulnerability of elements potentially involved in floods [38–40]. Earth observation based on remote sensing and geographical information systems (GIS) is essential for the implementation of quantitative and qualitative analysis in these fields [41–45]. In particular, multi-temporal analysis of data retrieved from historical maps and aerial or satellite imagery, along with information on past floods, is fundamental to depicting the landscape evolution over time and possible flood scenarios [20,46–54].

Numerous studies have investigated the relationship between urbanization and fluvial flood hazard [18,55–58]. Considering urban expansion as a key driver of floods, they mainly focused on the impact of urbanization on flood peaks [59,60], rainfall-runoff regime [61,62], flood frequency [63,64], and flood magnitude [65,66]. In contrast, as far as is known, flood hazard as driver of the urbanization process, namely, the impact of floods on urban expansion, has not been documented in the scientific literature. Other works focused on the identification of flood-prone areas with respect to diverse flood scenarios [39,67,68], while relatively few studies assessed urban expansion over time in floodable areas [69–75]. These last issues are very relevant. The former represents the basis for defining land use planning and land management policies. The latter reveals the evolution of vulnerable elements exposed to hydrogeomorphic hazards over time, suggesting a simultaneous change in hydrogeomorphic risk [76].

Urbanization and exposure of anthropogenic elements to hydrogeomorphic hazards, often associated with specific historical periods or flood-management interventions along riverbeds [77–79], have been largely assessed in terms of land-use and land-cover change at a large spatial scale [21,23,79–81]. Analyses focused on individual buildings, therefore avoiding aggregation of data, were generally implemented for flood modeling in urban environments [82–84] and for potential and post-event damage assessment [85–87]. Less commonly, some studies on the exposure of individual buildings to hydrogeomorphic hazards [38,88,89] and a few works investigating in detail the historical evolution of urban settlements in terms of building footprint over time [20,46,90], namely at the individual-building scale, were performed.

Against this background, this study investigates past fluvial floods, urban expansion, and building exposure to floods in the cities of Garessio, Ceva, and Clavesana, along the Tanaro River (NW Italy, Figure 1). (i) the historical analysis of flood events that occurred over the last centuries, (ii) the detailed reconstruction of the urbanization process since the second half of the 19th century in terms of building footprint, (iii) the assessment of flood-related elements [91] associated with two high-magnitude floods of the Tanaro River (5–6 November 1994 and 24–25 November 2016) that were taken as references, and (iv) the analysis of extent and construction period of buildings that were inundated during these reference floods were performed.

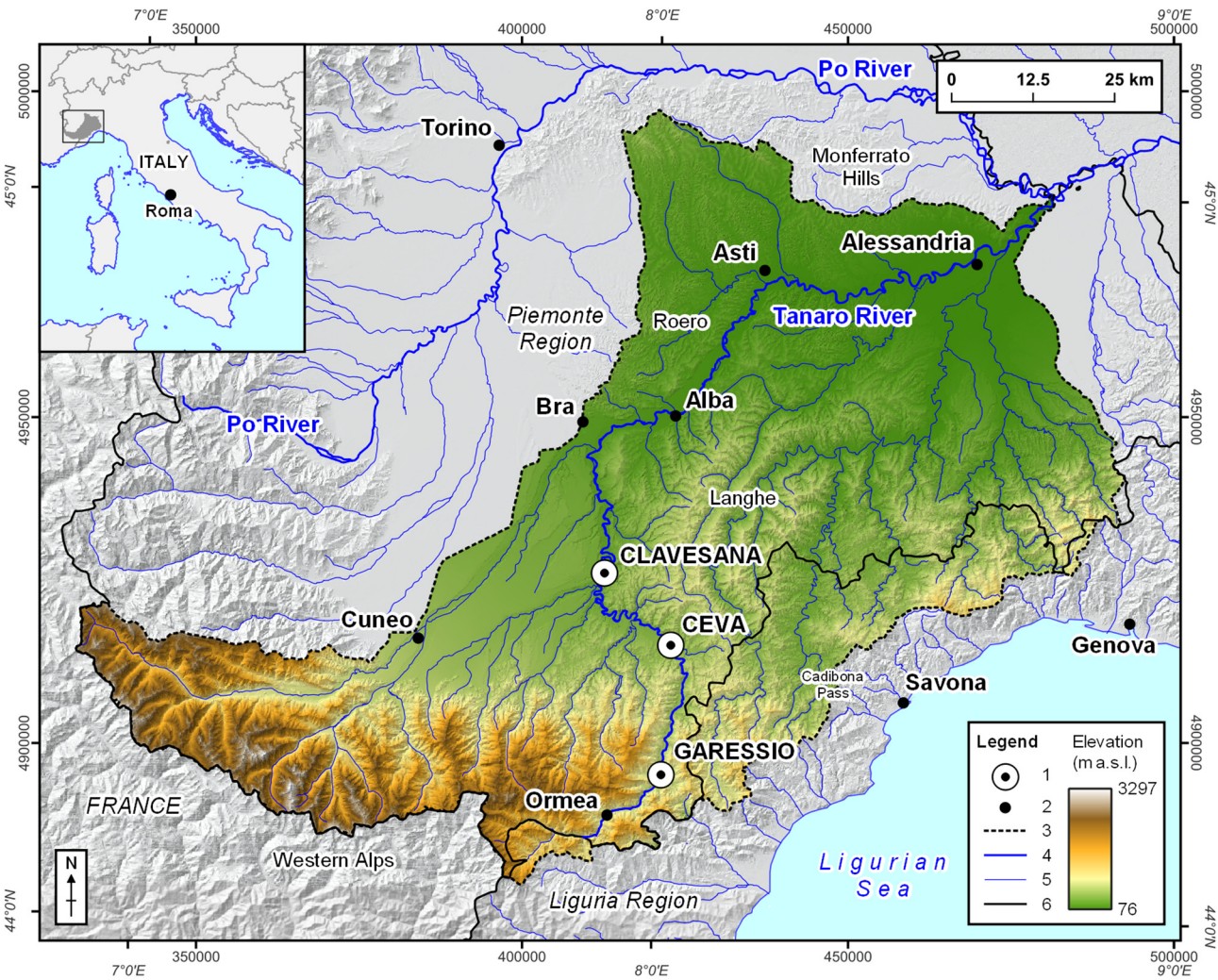

**Figure 1.** Location of the study areas. The colored hillshade corresponds to the Tanaro River catchment. 1: Investigated city; 2: Main city; 3: Divide of the Tanaro River catchment; 4: Tanaro and Po rivers; 5: Main hydrographic network; 6: Regional border.

The three cities considered are little villages straddling the river and located on a narrow valley floor bordered by slopes and terrace scarps. They were hit by the Tanaro floods several times over the last centuries and have experienced urban growth since the second half of the 19th century. Thus, Garessio, Ceva, and Clavesana are representative of several Italian urban settlements located in flood-related hazardous areas where the scarce extent of the valley floor represents a constraint for urban expansion. Moreover, the Tanaro River catchment represents a well-known open-air laboratory widely considered in research due to its proneness to damaging geo-hydrological events [20,73,92–98].

This research is based on historical sources, remote sensing data, and field evidence. Collection and review of historical data, photograph interpretation, GIS analysis, and field surveys were performed. The objectives of this study are: (i) to provide a methodological approach for further works focusing on urbanization and floodable areas; (ii) to introduce novel metrics for investigating urban expansion in terms of building footprint and building exposure to floods; (iii) to assess flood proneness, flood-related elements [91], urban expansion, and building exposure in Garessio, Ceva, and Clavesana; (iv) to emphasize the relevance of information on past floods and urbanization processes for land planning, land management, and hydrogeomorphic risk assessment; (v) to point up the need for land use planning for flood control to forbid new urban expansion in potentially floodable areas.

The outcomes represent an essential knowledge base to define effective and sustainable management measures to mitigate hydrogeomorphic risk.

## 2. Study Area

The study area corresponds to the urban area and surroundings of Garessio, Ceva, and Clavesana. These three urban settlements are located in the north-western part of Italy, within the Tanaro River catchment and along its main valley floor (Figure 1). The Tanaro River originates at the confluence between the Tanarello and Negrone Creeks in the Ligurian Alps, about 22 km upstream of Garessio, and flows north-eastwards to the Po River for some 276 km [99]. Its catchment has an overall area of approximately 8080 km$^2$ (11% of the Po River catchment), 82% of which is mountainous [99], and a total relief of 3221 m.

The Tanaro catchment can be divided into distinct orographic features: (i) the Alpine valleys west of the Cadibona Pass, represented by Maritime and Ligurian Alps, with narrow valleys, steep slopes, high reliefs, and, from a geological point of view, the Argentera Massif and the Provencal–Dauphinoise, Brianconnais, Prepiedmont and Piedmont units of the Palaeo–European continental margin; (ii) the Apennines valleys, est of the Cadibona Pass, still associated with Ligurian Alps from a geological point of view, with the aforementioned Brianconnais, Prepiedmont and Piedmont units, and the units of the Liguria–Piemonte oceanic domain, and showing narrow valleys, steep slopes, and lower reliefs; (iii) the hills of the Langhe, Roero and Monferrato, displaying both gentle slopes and deeply-incised valleys shaped into sedimentary units of the Tertiary Piedmont Basin (TPB) and Pliocene Succession; and (iv) the terraced floodplains of Savigliano and Alessandria presenting Quaternary deposits, upstream and downstream of the Langhe hills, respectively [100–107]. Rock outcroppings in the first two zones are antecedent to the Tertiary and mainly metamorphic and calcareous [96].

The basin is in a temperate climatic zone, with significant differences between its lower part (the Alessandria Plain) and the Alpine Ridge. The annual rainfall is extremely variable over the catchment, ranging between about 600 mm at the outlet and over 1800 mm in mountain areas [73,96,99]. The mean annual discharge of the Tanaro River is estimated to be about 116 m$^3$/s at the Po confluence [108]. The Tanaro River, with its catchment straddling the Alps and Apennines, presents a complex water regime characterized by a combination of features from the Alpine regime, typical of its tributaries from the Alps, and the Apennine regime, typical of its tributaries from the Apennines. In fact, approximately upstream of Ceva, it has an almost alpine regime, with major flows in late spring due to the snow melting and low discharges in summer and winter; downstream of Ceva, the river has two discharge peaks during the year, namely, one in late spring (due to snow melt) and one in autumn (due to rainfall), with the former slightly greater than the latter (typical feature of Alpine rivers), and two low flow periods, namely, one in summer and one in winter, with the former more marked than the latter (typical feature of Apennine rivers) [99].

The mountainous parts of the catchment are mostly covered by woods, while the hilly and lowland sectors mainly present agricultural areas. Urban settlements are widespread in lowland areas and concentrated over the valley floors in the mountainous and hilly sectors.

The Tanaro River catchment is notoriously prone to fluvial floods and slope instability processes [94,109,110]. Very severe and damaging geo-hydrological events triggered by both long-lasting and extremely brief precipitations have occurred in the past; recently, three major events occurred in 1994, 2016, and 2020, causing relevant damage due to flooding, geomorphic processes related to rivers and runoff, and landslides.

Garessio (urban settlement area: 2.2 km$^2$; municipality area: 131 km$^2$; inhabitants: c.a. 2900), Ceva (urban settlement area: 2 km$^2$; municipality area: 43 km$^2$; inhabitants: c.a. 5750), and Clavesana (urban settlement area: 0.3 km$^2$; municipality area: 17 km$^2$; inhabitants: c.a. 800) are located in the upper part of the Tanaro catchment, straddling

the Tanaro River (Figures 1 and 2). The origin of the first settlements where these cities developed dates back to the pre-Roman period. Their growth in terms of built-up surface was relatively scarce up until the mid-19th century.

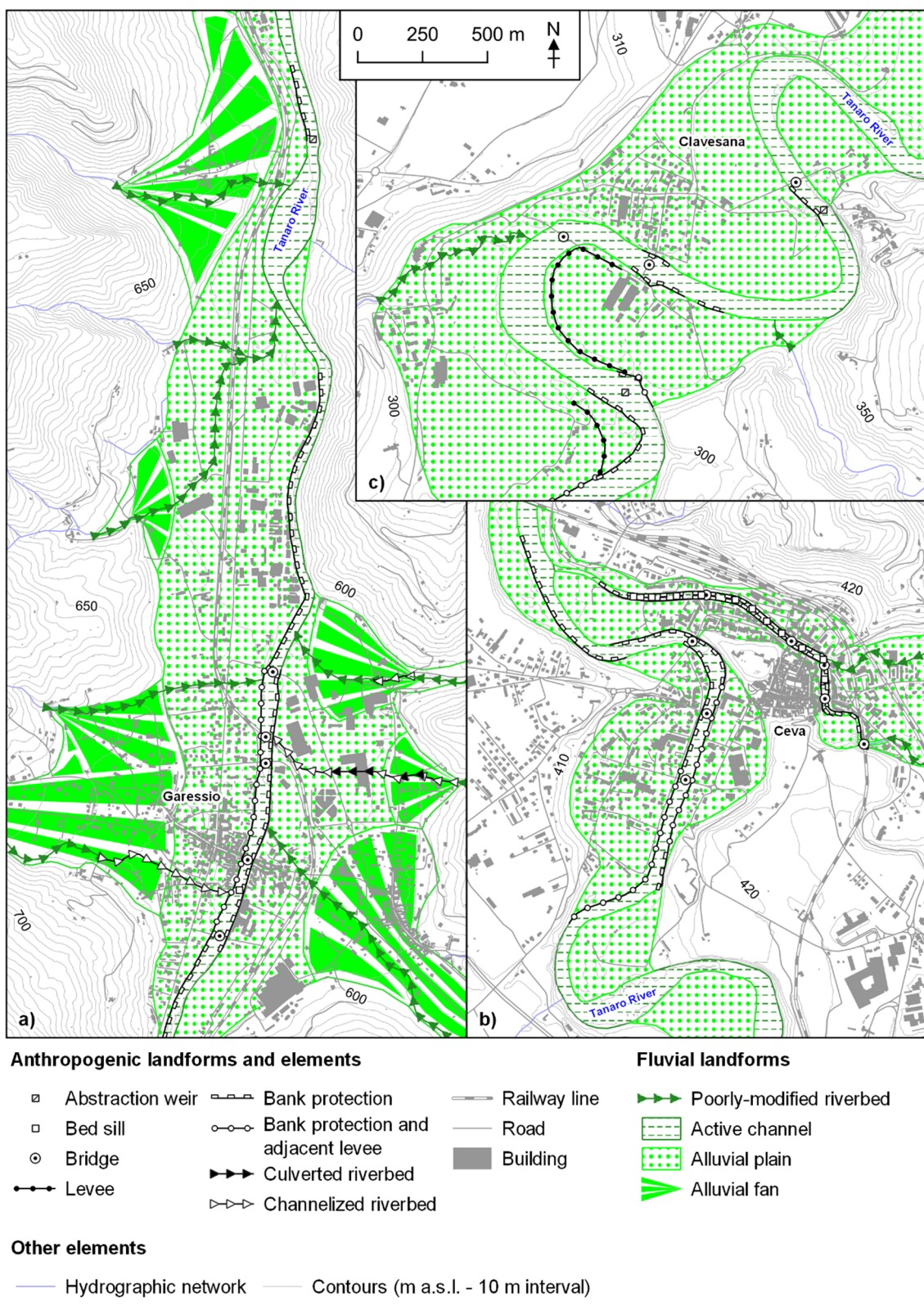

**Figure 2.** Geomorphological features of (**a**) Garessio, (**b**) Ceva, and (**c**) Clavesana valley floor.

Garessio is in the Ligurian Alps, in a 600–1100 m wide valley floor bordered by high and steep slopes constituted of Briançonnais metasedimentary units [107], and presenting a recent alluvial plain shaped by the Tanaro River and large alluvial fans modeled by the right and left tributaries (Figure 2a). Here, the Tanaro riverbed has coarse-grained sediment and displays a single-thread channel severely conditioned by anthropogenic structures for erosion and flood defense. Most of the valley floor is covered by structures and infrastructure. Downstream of the city center, there is a very large industrial plant close to the riverbed; moreover, there are some cultivated areas around the urban area, mainly for forage provision.

Ceva is located where the Alpine valley ends and the landscape becomes hilly with lower and gentler slopes, wider valleys, and high fluvial terraces bordering deeply incised valley floors. The city spreads over two main levels: one corresponding to the fluvial terrace covered by Pleistocene alluvial deposits and the other, 35–40 m lower, corresponding to the modern valley floor entrenched between fluvial terrace scarps constituted of sedimentary rocks belonging to the TPB succession (Figure 2b) [107,111]. The Tanaro River is single-thread with coarse-grained sediment, sinuous upstream and downstream of the city, and straight and severely channelized within the urban reach. The Cevetta Creek borders the historical village to the east and north; its riverbed displays numerous bed sills and joins the Tanaro in the downstream-most part of Ceva.

Clavesana is at the limit between the Savigliano terraced floodplain and the Langhe hills and is composed of two distinct villages: (i) the old town, which is located on the right side of the Tanaro Valley 20–60 m high above the river, and (ii) the modern town, which is located over the valley floor on the left side (Figure 2c). The valley floor is entrenched between steep slopes and scarps of fluvial terraces composed of sedimentary rocks belonging to the TPB succession [107]. The wide fluvial terrace on the left side of the valley is covered by Pleistocene alluvial deposits, whereas the valley floor presents recent alluvial deposits. The Tanaro riverbed is single-thread with coarse-grained sediment and presents meanders largely confined by terrace scarps and hill slopes.

## 3. Materials and Methods

The overall methodological framework implemented to perform this research is reported in Figure 3.

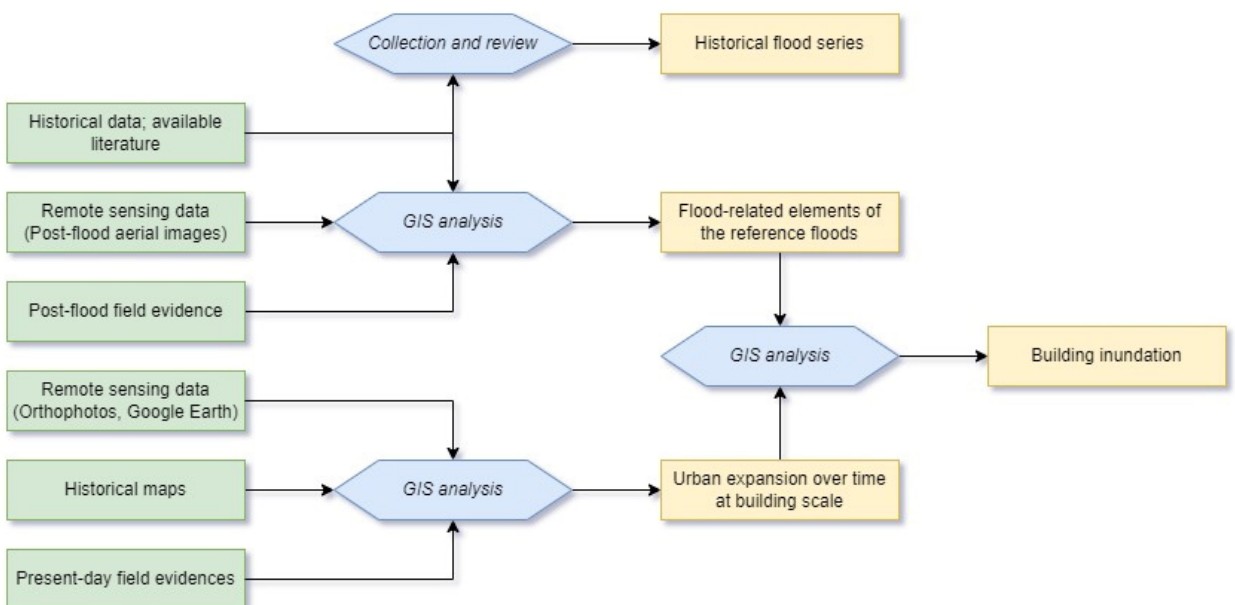

**Figure 3.** Schematization of the methodological framework. Rectangle indicates input (greenish) or output (yellowish) data; hexagon indicates process.

The geographical data listed in Table 1 were imported into a GIS environment through highly accurate georeferencing and considered a base for mapping and dating buildings constituting the urban fabric of Garessio, Ceva, and Clavesana. Buildings were both retrieved as polygons from the "Building vector layer of the regional database" (Table 1) and manually digitized on maps and images (Table 1) at a constant scale of 1:2000 by a single operator. The detailed and correct representation of buildings on historical maps was assumed. Moreover, the availability of only these data (Table 1) made possible uncertainties associated with the combination of different data sources to be overlooked. However, according to historical documents and field evidence, relevant uncertainties related to building extent and location can be excluded. Then, polygons representing the building footprint were classified according to their construction period. According to historical data availability, the following time spans were considered: before 1852; 1852–1933, 1933–1994, 1994–2016 for Garessio; before 1852; 1852–1930, 1930–1994, 1994–2016 for Ceva; before 1870; 1870–1933, 1933–1994, 1994–2016 for Clavesana. Field surveys were conducted to verify the present-day urban setting and identify the main geomorphological features of the Tanaro River valley floor. According to field evidence, geological data (see references in the previous section), and a shaded relief map, two main landscape units were mapped, namely the valley floor and alluvial fan. These were considered to further investigate urban expansion, focusing on flood-prone areas. The shaded-relief map was derived from a freely available high-resolution digital terrain model (5-m-cell size) produced with LIDAR data from a 2009 survey (property of the Piemonte Region).

**Table 1.** Summary of geographical data sources considered in this research. B/W: black-white; C: colored; WFS: Web Feature Service; WMS: Web Map Service.

| Year | City | Datum Description | Scale | Data Source for GIS |
|---|---|---|---|---|
| 1852 | Garessio, Ceva | Historical map, 'Gran Carta degli Stati Sardi in Terraferma' | 1:50,000 | Digital image ** |
| 1870 | Clavesana | Historical map, 'Gran Carta degli Stati Sardi in Terraferma' | 1:50,000 | Digital image ** |
| 1930 | Ceva | Historical map of the Military Geographic Institute of Italy | 1:25,000 | Digital image ** |
| 1933 | Garessio, Clavesana | Historical map of the Military Geographic Institute of Italy | 1:25,000 | Digital image ** |
| 1991 * | Garessio, Ceva, Clavesana | Orthophotos (B/W) | | Regional Geoportal WMS |
| 1991 * | Garessio, Ceva, Clavesana | Regional map 'Carta Tecnica Regionale' | 1:10,000 | Digital image ** |
| 1994 | Garessio, Ceva, Clavesana | Aerial photographs (B/W) | 1:10,000 | Digital image ** |
| 2015 * | Garessio, Ceva, Clavesana | Orthophotos (C) | | Regional Geoportal WMS |
| 2016 | Garessio, Ceva, Clavesana | Regional map, 'Banca Dati Territoriale di Riferimento degli Enti—BDTRE' | 1:10,000 | Regional Geoportal WMS |
| 2016 | Clavesana, Ceva | Orthophotos (C) | | Regional Geoportal WMS |
| 2017 * | Garessio, Ceva, Clavesana | Building vector layer of the regional database | 1:10,000 | Regional Geoportal WFS |
| 2017 * | Garessio, Ceva, Clavesana | Google Earth images (C) | | Google Earth |

*: datum considered to improve mapping of buildings in 1994 and 2016; **: datum retrieved from the archive of the National Research Council—Research Institute for Geo-Hydrological Protection.

A set of GIS-based metrics was introduced to quantitatively outline the urban growth of the investigated cities at the individual-building scale, thus focusing on the building footprint. Each polygon representing building footprint was considered, and (i) the overall area of buildings, i.e., the aforementioned building footprint, that were built (BA), (ii) the

rate of building expansion (BE Rate), i.e., the ratio between BA and the period duration in years, (iii) the overall area of buildings that were built over the valley floor (BAvf), (iv) the rate of building expansion over the valley floor (BEvf Rate), i.e., the ratio between BAvf and the period duration in years, and (v) the percentage of BAvf out of BA (BAvf Ratio), namely BAvf/BA × 100, were computed with reference to every construction period. Moreover, the cumulative BA (CBA), i.e., the progressive sum of BA, and the cumulative BAvf (CBAvf), i.e., the progressive sum of BAvf, were calculated.

A very large activity of research and collection of data concerning past floods was carried out to depict the flood history in Garessio, Ceva, and Clavesana and their flood-proneness, and to identify the most representative flood events of the Tanaro River that hit the three cities investigated. A wide set of sources was considered, including scientific papers, manuscripts, unpublished technical reports, maps, projects, newspapers, photographs, and interviews with elderly people and local inhabitants (Figure 4). Most information was retrieved from (i) the state office archives (i.e., Ministry of Public Works archive, state archive, Hydrographical Office for the Po River archive, etc.), where many unpublished reports on past floods are preserved, (ii) municipal libraries, which provide hold papers, technical and historical books, and sometimes local newspaper articles, and (iii) municipal archives, which contain detailed information about past floods and references to the inundated areas, casualties, and damage at a local scale. All documents were examined, validated, and imported into a database. Flood events were classified into four classes (low, medium, high, and very high damage) according to the level of economic losses they caused. The extent of damage in monetary terms, especially damage to public works, related to each flood was estimated on the basis of data from archival documents and by means of currency converters [94].

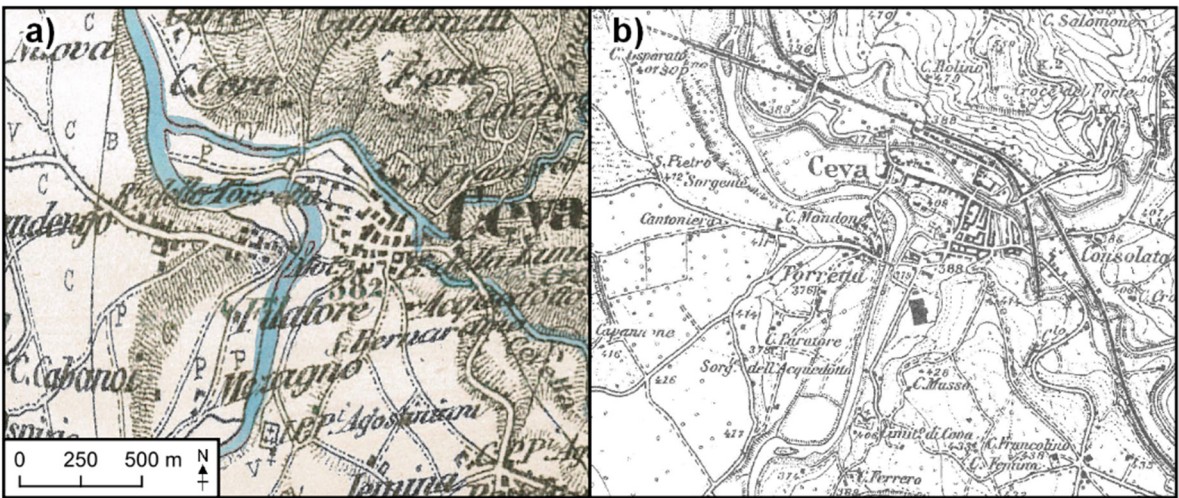

**Figure 4.** Example of historical maps that were used in this research to investigate urban evolution. Ceva in (**a**) 1852 and (**b**) 1930.

The high-magnitude events that occurred on 5–6 November 1994 and 24–25 November 2016 were taken as references, and their flood-related elements [91] were analyzed in detail. Flood-induced ground effects and flood-water dynamics (i.e., water depth, direction of flows and main flows, and flooded area) [91] were outlined by means of original field surveys conducted immediately after the events, aerial image photointerpretation performed in a GIS environment (Table 1), and data from the archives of the Environment Protection Agency of the Piemonte Region.

Ultimately, the area and construction period of buildings that were inundated during the two aforementioned major floods (namely, in 1994 and 2016) and the impact of these past flood events on urban development over the valley floor were investigated. An overlay procedure based on vector layers of building footprint and flooded area was implemented

to identify the flooded buildings. A set of GIS-based metrics was defined to assess them quantitatively: (i) the Area of Flooded Buildings (AFB) referred to construction periods; (ii) the Cumulative AFB (CAFB), i.e., the progressive sum of AFB referring to construction periods; (iii) the ratio between the AFB and the area of buildings that were built over the valley floor, as AFB/BAvf × 100, referring to both the total values of AFB and BAvf (AFBvf) and to their values associated with each construction period (AFBvf_cp); (iv) the percentage value of the total AFB referred to each construction period (AFBcp). In Garessio, BAvf, CBAvf, BEvf Rate, AFBvf, and AFBvf_cp considered buildings that were built over both the alluvial fans and the valley floor.

All GIS analyses were performed using the free and open-source software QGIS 3.22.

## 4. Results

### 4.1. Urban Expansion

The extent of building surface in Garessio, Ceva, and Clavesana over the time span investigated is reported in Table 2.

**Table 2.** Urban expansion over time in terms of building area.

| City | Construction Period | BA | | CBA | | BE Rate | BAvf | | CBAvf | | BEvf Rate | BAvf Ratio |
|------|---------------------|------|------|------|------|---------|------|------|-------|-------|-----------|------------|
| | | (ha) | (%) | (ha) | (%) | (ha/yr) | (ha) | (%) | (ha) | (%) | (ha/yr) | (%) |
| Garessio * | Before 1852 | 5.0 | 13.8 | 5.0 | 13.8 | | 4.2 | 14.1 | 4.2 | 14.1 | | 83.1 |
| | 1852–1933 | 5.2 | 14.1 | 10.2 | 27.9 | 0.06 | 4.7 | 16.0 | 8.9 | 30.1 | 0.06 | 91.7 |
| | 1933–1994 | 18.4 | 50.1 | 28.6 | 78.0 | 0.30 | 15.9 | 53.6 | 24.8 | 83.7 | 0.26 | 86.7 |
| | 1994–2016 | 8.1 | 22.0 | 36.6 | 100.0 | 0.37 | 4.8 | 16.3 | 29.7 | 100.0 | 0.22 | 60.0 |
| | Total | 36.6 | | | | | 29.7 | | | | | 81.0 |
| Ceva | Before 1852 | 4.6 | 11.2 | 4.6 | 11.2 | | 1.0 | 7.9 | 1.0 | 7.9 | | 20.9 |
| | 1852–1930 | 5.2 | 12.4 | 9.8 | 23.6 | 0.07 | 3.3 | 26.7 | 4.3 | 34.5 | 0.04 | 63.7 |
| | 1930–1994 | 21.3 | 51.4 | 31.1 | 75.0 | 0.33 | 7.1 | 57.5 | 11.3 | 92.1 | 0.11 | 33.2 |
| | 1994–2016 | 10.4 | 25.0 | 41.5 | 100.0 | 0.47 | 1.0 | 7.9 | 12.3 | 100.0 | 0.04 | 9.4 |
| | Total | 41.5 | | | | | 12.3 | | | | | 29.7 |
| Clavesana | Before 1870 | 1.3 | 14.0 | 1.3 | 14.0 | | 0.4 | 8.0 | 0.4 | 8.0 | | 32.2 |
| | 1870–1933 | 2.3 | 24.5 | 3.6 | 38.5 | 0.04 | 1.6 | 30.5 | 2.0 | 38.5 | 0.03 | 69.6 |
| | 1933–1994 | 3.9 | 41.9 | 7.5 | 80.4 | 0.06 | 3.0 | 56.7 | 5.0 | 95.2 | 0.05 | 75.9 |
| | 1994–2016 | 1.8 | 19.6 | 9.4 | 100.0 | 0.08 | 0.3 | 4.8 | 5.3 | 100.0 | 0.01 | 13.6 |
| | Total | 9.4 | | | | | 5.3 | | | | | 56.0 |

* As for Garessio, metrics referring to the valley floor (i.e., BAvf, CBAvf, BEvf Rate, and BAvf Ratio) were computed considering buildings that were built over both the alluvial fans and the valley floor.

As for Garessio, in 1852, two distinct villages existed over the main valley floor, namely, Borgo Ponte and Borgo Maggiore (Figure 5a). The former, straddling the Tanaro River, presented two inhabited nuclei joined by a bridge; the latter, east of Borgo Ponte, was located at the outlet of the Rocca Bianca Valley. A slight increase in built-up area was documented in the period 1852–1933, associated with the expansion of Borgo Ponte and the establishment of industrial plants downstream of the village. The most intense urban growth was observed during the 20th century, particularly from the beginning of the 1950s [109], when the overall area of buildings tripled (Table 2). Large surfaces were involved in this process over both the valley floor and the alluvial fans around Borgo Ponte and downstream of the village center close to the river (Figure 5a). Half of the overall building area is dated back to the period 1933–1994, and 87% of the surface built-up in 1933–1994 is located in the valley floor and on alluvial fans (Table 2). Between 1994 and

2016, a slight increase in building area was registered again, mainly related to industrial sheds.

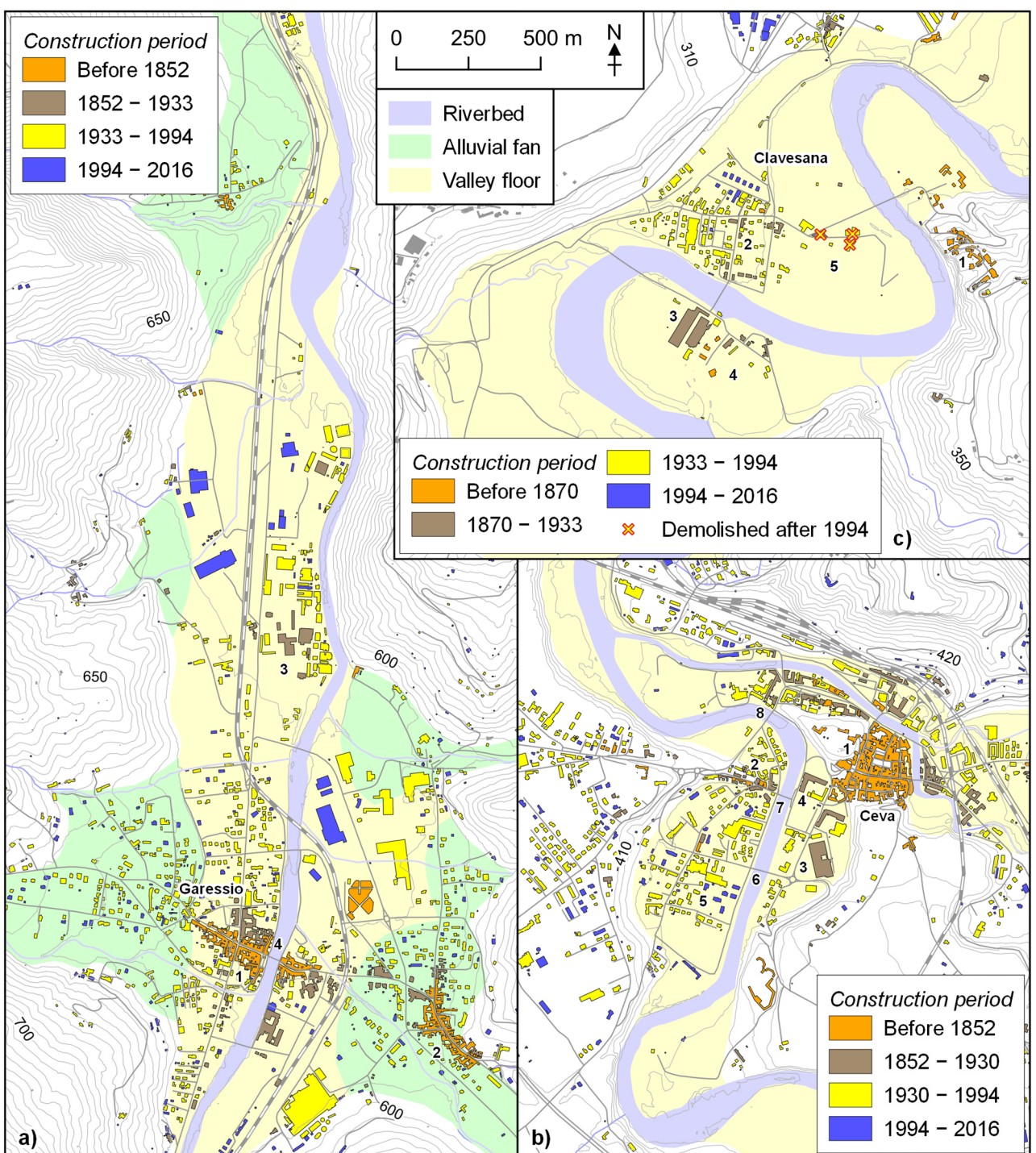

**Figure 5.** Schemes follow the same formatting. Urban expansion over time in terms of buildings. (**a**) Garessio. 1: Borgo Ponte quarter; 2: Borgo Maggiore quarter and Rocca Bianca Valley outlet; 3: Huvepharma company (former Lepetit); 4: Odasso Bridge. (**b**) Ceva. 1: Historical center; 2: Torretta quarter; 3: Broglio quarter; 4: Galliano Barrack; 5: Polveriera Street; 6: Provincial Road 225 bridge; 7: Footbridge; 8: Catalana bridge. (**c**) Clavesana. 1: Historical center; 2: Madonna della Neve quarter; 3: Cotton mill; 4: Gerino quarter; 5: La Generala quarter.

Up until the mid-19th century, the urban settlement of Ceva remained strictly confined around its castle and cathedral, which are located about 30 m higher than the surrounding areas, just upstream of the confluence between the Tanaro River and the Cevetta Stream (Figure 5b). Subsequently, several buildings, such as barracks, industrial sheds, nursery schools, and mills, arose between the 1850s and the 1890s over the valley floor in the Broglio quarter, and other buildings were built in the first decades of the twentieth century along the Cevetta and in the Broglio and Torretta quarters [109]. The major urban expansion occurred between 1930 and 1994. One third of the built-up surface referred to in this period was included within the valley floor (Table 2). According to historical sources [109], most of the building area increase dates back to the 1960s, when residential buildings, an oratory, a football field, and a school were built on the right bank and several houses and businesses on the left bank; similarly, several houses were built along the Cevetta. Moreover, from the 1970s to 1994, the urban expansion of Ceva continued, affecting almost completely the valley floor. After 1994, one fourth of the overall BA was built up, partly (i.e., 9%) over the valley floor (Table 2, Figure 5b).

In Clavesana, the old town remained substantially unchanged over time, while the modern village experienced a progressive increase in building area (Figure 5c). Urban expansion in the period 1870–1933 was limited to some farms on both sides of the valley and a large cotton mill over the valley floor in a meander lobe. Most of the building surface dates back to the period 1933–1994 (Table 2); in the mid-20th century, a sports facility was realized adjacent to the riverbed on the aforementioned meander lobe [94]. Considering the periods 1870–1933 and 1933–1994, 70% and 76% of urban expansion, respectively, occurred over the valley floor (Table 2). After 1994, some buildings were built in the northern part of the modern town, and three constructions severely damaged by the 1994 flood were demolished in La Generala quarter (Figure 5c).

Thus, urban expansion occurred in all three cities over the time span considered, especially from the 1930s to 1994 (Table 2). This process affected both the valley floor and adjacent areas. Garessio and Ceva experienced quite similar BA in each period, with differences never higher than 3.0 ha (in the 1930s–1994) and values ranging from 4.6 ha to 21.3 ha. Conversely, urban expansion in Clavesana was markedly lower, with a maximum BA of 3.9 ha (in the 1933–1994). Focusing on the valley bottom, different similarities were observed (Table 2). The BAvf values referred to the first period considered were similar and very low in Ceva and Clavesana and more than four times higher in Garessio. The highest (lowest) BAvf values were observed in Garessio (Clavesana) for each period. Values ranging between 53.6% (Garessio) and 57.5% (Ceva) of the total BAvf in 2016 were documented with reference to the construction period of the 1930s–1994. It is noteworthy that an increase in BAvf lower than 1.0 ha in Ceva and Clavesana and by 4.8 ha in Garessio also occurred in the most recent period considered. In all cities, the highest rate of BA variation was observed during the most recent period and between the 1930s and 1994. In 1994, the building surface in Garessio, Ceva, and Clavesana was about 75–80% (84–95%) of the total BA (CBAvf) reached in 2016, thus most of the buildings were already realized. In 2016, 81% of the total area occupied by buildings in Garessio was included within the valley floor. This value was 30% in Ceva and 56% in Clavesana.

### 4.2. Historical Floods of the Tanaro River

The Tanaro River has a very long history of flood events; the first available documents testifying floods with damage date back to the 13th century [90,109]. In the 19th century, 17 floods with damage occurred along the main valley floor; the most catastrophic event was in May 1879 [90]. In the 20th century, there were 25 floods with damage, and the most serious occurred on 5–6 November 1994. This hit 38 inhabited centers, causing 44 fatalities, 2000 displaced persons, and over 10 billion USD in damages [94]. Percentages of flooded urban areas ranged from 5% to 100%, with an average value of 30% [112]. After 2000, two high-magnitude floods were registered in the upper part of the Tanaro Valley, the former

on 24–25 November 2016 and the latter on 2–3 October 2020 [109]. These caused serious damage over the valley floor, fortunately without casualties.

As for Garessio, Ceva, and Clavesana, the series of damaging floods of the Tanaro River that occurred from the early 17th century is reported in Table 3.

**Table 3.** List of historical floods that hit Garessio, Ceva, and Clavesana causing damage. The damage level is reported: low, medium, high, and very high.

| Date | Garessio | Ceva | Clavesana | Date | Garessio | Ceva | Clavesana |
|---|---|---|---|---|---|---|---|
| 13–14 January 1610 | | Very high | | 18 September 1907 | High | | |
| 6 November 1612 | Low | | | 28 September 1907 | | Medium | |
| 18 August 1665 | Medium | | | 3 October 1907 | Medium | | |
| 21 April 1716 | High | | | 16 October 1907 | | Medium | |
| 25 July 1716 | High | | | 3 March 1914 | Medium | | |
| 30 October 1744 | High | | | 9 September 1914 | | Medium | |
| 27 July 1747 | High | | | 14 September 1914 | High | | |
| 9–13 October 1791 | High | | | 23 September 1920 | Low | | |
| 19 November 1791 | Medium | | | 15–16 May 1926 | | Medium | High |
| 1 October 1792 | High | | | 21–22 November 1926 | | | Low |
| 19 April 1796 | | High | | 11 November 1951 | | | High |
| 28 July 1798 | Medium | | | 13–16 June 1957 | | | Medium |
| 26 May 1879 | | | Medium | 18 December 1960 | | | Medium |
| 3 November 1843 | | High | | 8 November 1962 | | High | High |
| 17 October 1885 | | Medium | | 3 November 1968 | | | Medium |
| 21 November 1885 | | Medium | | 19 November 1970 | Low | | |
| October–November * 1885 | | | Medium | 14–15 October 1976 | Medium | | |
| 10–11 November 1886 | Very high | Very high | | 5–6 November 1994 | Very high | Very high | Very high |
| 18 May 1890 | | Medium | | 24–25 November 2016 | Very high | Very high | Very high |
| 28 October 1896 | Low | | | 2–3 October 2020 | Very high | Very high | Very high |
| 1 November 1906 | | Medium | | | | | |

* Date not known exactly.

Garessio was hit 22 times starting from 1612 to 2020; this means on average one flood event every 18.5 years. By narrowing the period considered, it is noteworthy that from 1886 to 2020 there were 12 events, which lowers the average frequency to one event every 11 years. This value changes to one every 8.6 years, considering the time span of 1994–2020. Ceva experienced 17 damaging floods between 1610 and 2020, namely, on average, one event every 24.1 years. In the period 1885 to 2020 the city suffered 14 floods, which lowers the average to one event every 9.6 years (Figure 6). Clavesana was hit 12 times from 1879 to 2020, namely, on average once every 11.7 years. Considering the period 1951–2020 the frequency increased to one event every 8.6 years, as 8 events were documented. Between 1994 and 2020, three floods causing very high damage occurred in all three cities, i.e., on average, one every 8.7 years.

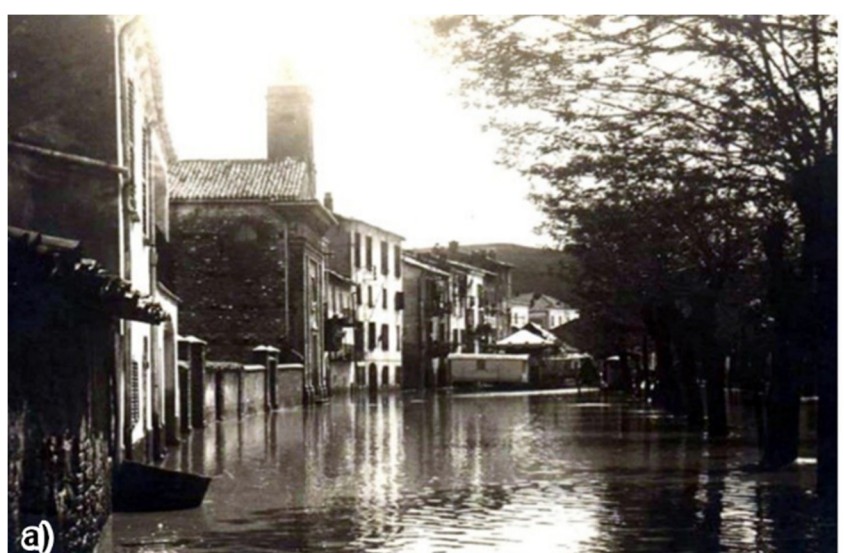
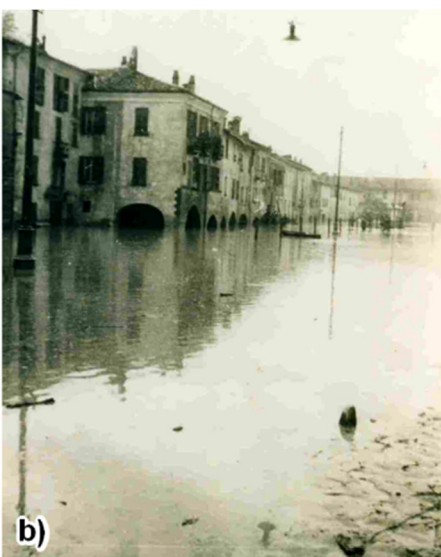

**Figure 6.** Historical floods in Ceva. (**a**) 16 May 1926. (**b**) 28 October 1942; in this event, flood water depth rose up to 2 m above the street level. Photos from the Municipal Archive of Ceva.

### 4.3. The 1994 and 2016 High-Magnitude Floods

The 5–6 November 1994 and 24–25 November 2016 high-magnitude floods were triggered by intense rainfall related to a large-scale atmospheric setting presenting a large North Atlantic low-pressure area deepening from the British Isles towards the North African coast and a robust high-pressure field over the eastern Mediterranean areas [94,113,114]. However, substantial differences between them lie in (i) the duration of rainfall before the paroxysmal flood event, (ii) the western Mediterranean Sea surface and air temperature, and (iii) the spatial distribution of precipitations. Unlike in 1994, in 2016 prolonged rainfalls occurred in the days before the flood; furthermore, in 1994 the air column was warmer than in 2016, whereas the surficial temperature of the sea was higher than the autumn average values in 2016 and in accordance with them in 1994 [113]. These elements were associated with a higher gradient of pressure close to the Western Alps in 2016 than in 1994. The average value of rainfall referred to the whole Tanaro catchment area was 230 mm and 357 mm for the periods 3–7 November 1994 and 21–25 November 2016, respectively [113]. In the upper part of the Tanaro catchment, it rained more in 2016 than in 1994 (e.g., in Ormea, 647 mm in the period 20–25 November 2016 and 376 mm in the period 2–7 November 1994); on the contrary, the downstream hilly (i.e., Langhe, Monferrato) and floodplain areas received a lower amount of rain in 2016 (e.g., in Alba, 148 mm and 251 mm, referring to the aforementioned time spans) [115].

Very relevant flood peaks on the Tanaro River corresponded to these rainfalls. In Alba, the maximum in-channel water level was 6.74 m (4200 $m^3/s$) and 6.14 m (3400 $m^3/s$) in 1994 and 2016, respectively. However, the former corresponds to the maximum value registered before the gauging station broke down. Downstream of Alessandria, the maximum level was 8.48 m (namely, 4400 $m^3/s$) in 1994 and 7.72 m (namely, 3720 $m^3/s$) in 2016; the 1994 value was the highest ever recorded since 1904, the first year of the time series [115]. While an only peak was overall registered in 1994, two peaks 10–12 h apart occurred upstream of Asti in 2016 due to the contribution of tributaries; downstream of Asti they joined, and an only major peak propagated up to the Po River.

Garessio, Ceva, and Clavesana were seriously affected by the 1994 and 2016 floods on November 5 and 24, respectively, with dramatic consequences (Figure 7).

**1994 flood**

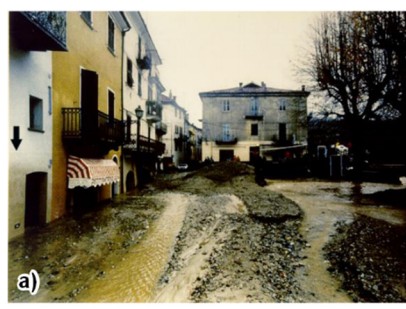
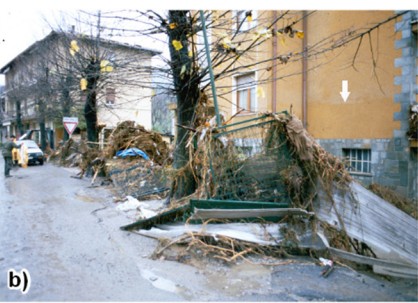
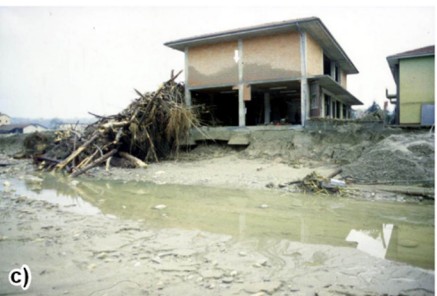

**2016 flood**

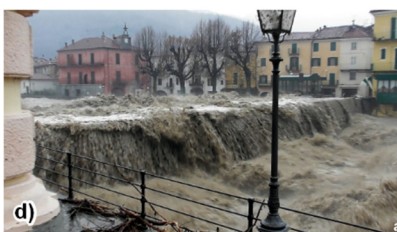
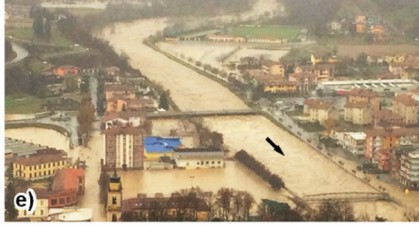
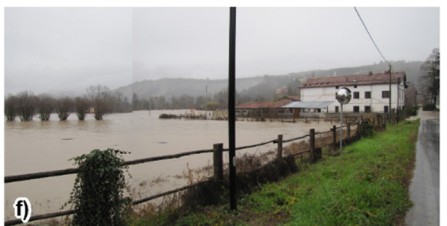

**Figure 7.** (**a**) Garessio after the 1994 flood. The black arrow indicates the level reached by floodwater during the night (2.5 m above ground level); the square close to the Odasso Bridge was covered by a layer of sediments 80 cm thick. (**b**) Ceva after the 1994 flood. The white arrow indicates the maximum height reached by floodwater. (**c**) Clavesana after the 1994 flood. The flood caused the removal of the house's perimeter walls and the deposit of a large amount of floating material (coarse woody debris and shrubs). Moreover, surficial erosion processes resulted in a 2 m lowering of the ground surface. The white arrow indicates the maximum height reached by floodwater during the night between 5 and 6 November 1994. (**d**) Garessio and the Odasso Bridge overtopping during the 24–25 November 2016 night. (**e**) Ceva during the 2016 flood. The black arrow indicates the flow direction. On the left (right bank), the Broglio quarter. The kindergarten (blue roof) is completely flooded, as are the other buildings nearby. (**f**) Clavesana during the 2016 flood. Large agricultural areas were flooded, and buildings remained isolated. Photos by: (**a–c,f**) F. Luino; (**d**) A. Acquarone; (**e**) G. Galliano.

As for the 1994 event, Garessio was flooded with water heights up to 2.5 m above ground level, and the Odasso historic bridge was surmounted (Figures 7a and 8a). A total area of 105.4 ha was flooded. Several houses and at least thirty commercial activities were filled with muddy waters and floating materials. The school, the Ceva–Ormea railway line, and large industrial structures downstream of the village were seriously damaged. Furthermore, a bridge in the town center collapsed. The maximum discharge in Ceva was estimated to be about 1300 m$^3$/s for the Tanaro River and 300 m$^3$/s for the Cevetta Stream [109]. The valley floor was completely inundated (104.0 ha) with maximum water levels up to 4 m [114,116] (Figure 8b).

Several buildings were severely damaged by flood water and transported materials, especially at the upstream limit of the village along the Tanaro and at the river bend located at the foothills of the historical center (Figure 7b). All bridges were seriously damaged due to pier scouring and the impact of floating tree trunks. Diffuse bank erosions, causing the collapse of bank protections, were triggered along with surficial erosion and sediment deposition over the valley floor, particularly in agricultural areas [114]. The flood moved downstream in a SW-NE direction at Clavesana, flooding the valley floor and completely ignoring meanders, and involved buildings and structures located on meander lobes, causing two fatalities (Figures 7c and 8c). The abstraction weir of the former cotton mill, located upstream of Clavesana, trapped a large quantity of floating material, which diverted flood water towards Gerino and La Generala (Figure 9a). As a result, an alluvial gully [91,117] 50 m width, 400 m length, and more than 1 m depth was formed (Figure 9b). Downstream of this novel channel, at La Generala, water levels up to 3.4 m were registered,

two buildings were completely burned down, and many others were seriously damaged (Figure 7c). Thus, large areas over the valley floor were flooded (194.5 ha) and affected by surficial erosion and sediment deposition. Widespread bank failures occurred.

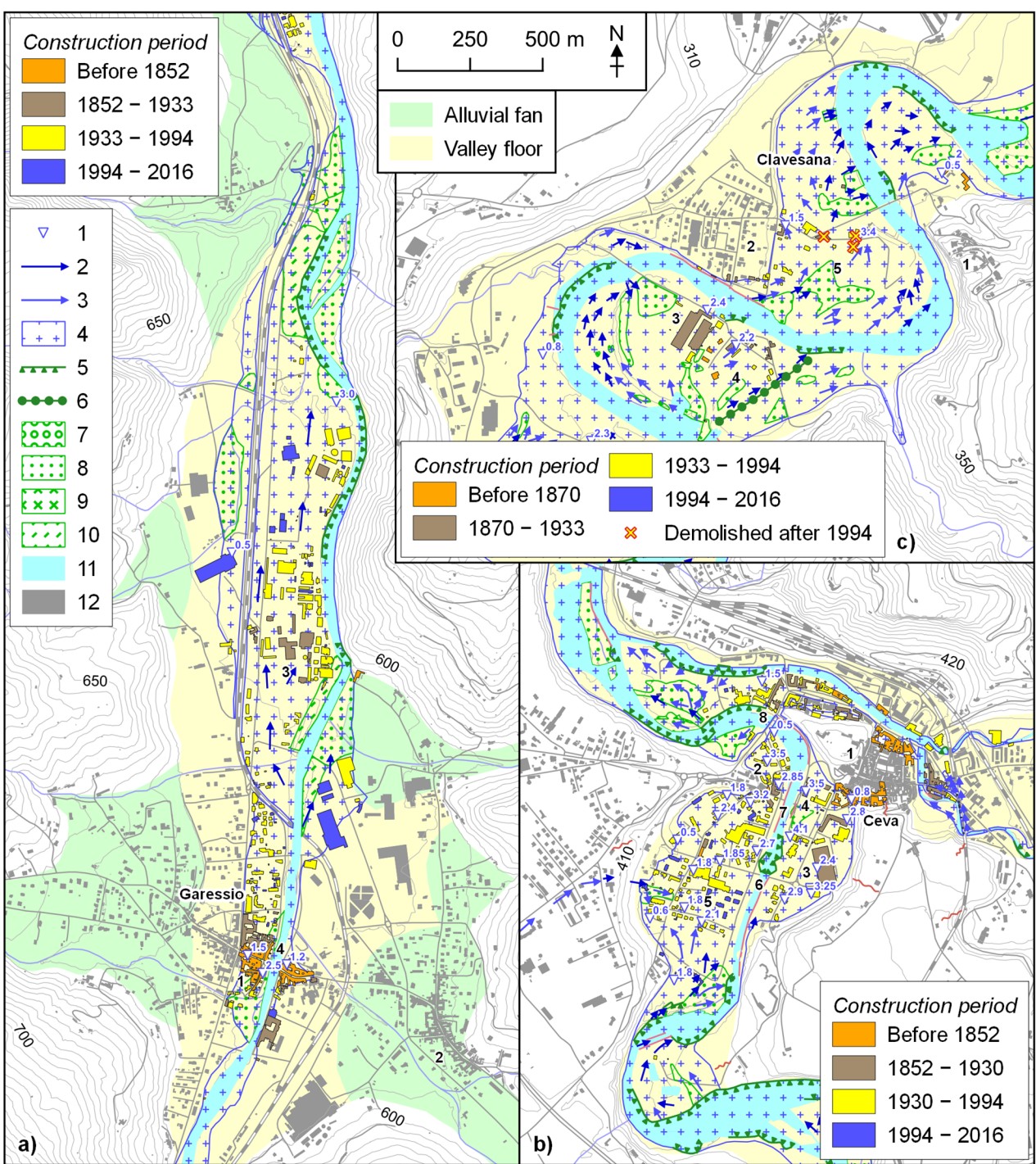

**Figure 8.** The 1994 flood. Flood-related elements and construction period of inundated buildings. (**a**) Garessio. (**b**) Ceva. (**c**) Clavesana. Legend for flood-related elements: 1: Water depth (m); 2: Direction of main flood flows; 3: Direction of flood flows; 4: Flooded area; 5: Bank erosion; 6: Flood channel; 7: Overbank deposition of coarse sediments; 8: Overbank deposition of fine sediments; 9: Overbank deposition of unsorted sediments; 10: Surficial erosion; 11: Riverbed after the flood; 12: non-flooded buildings. Refer to Figure 5 caption for the meaning of the numbers on the map.

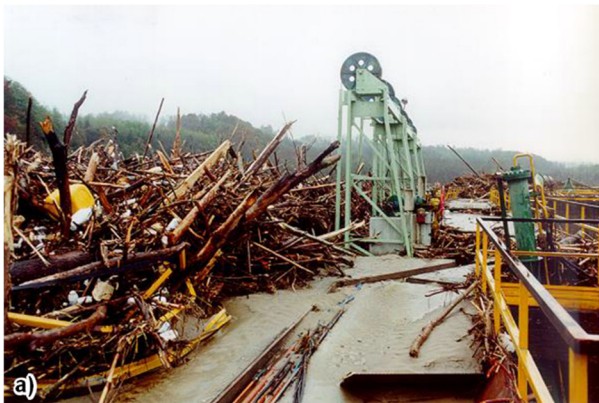 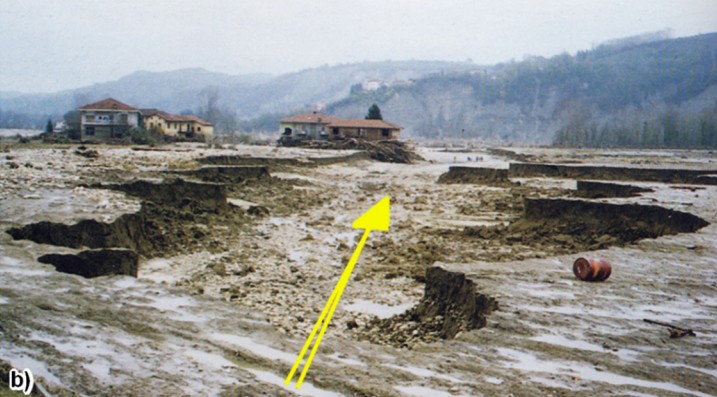

**Figure 9.** (**a**) Floating materials trapped by the abstraction weir upstream of Clavesana during the 1994 flood. (**b**) Alluvial gully shaped by floodwater at Gerino during the 1994 flood. The yellow arrow indicates the flow direction. Photos by: (**a**) F. Luino; (**b**) G. Susella.

During the 2016 flood, Garessio registered a maximum in-channel water level of 5.19 m, corresponding to about 830 m$^3$/s discharge (Figures 7d and 10a). This was approximately 1 m lower than the 1994 value, according to flood marks [115]. Similarly, a bit lower water level than in 1994 was registered in Ceva, partly due to the absence of the Cevetta contribution to the flood (Figure 10b). However, several damages to buildings, structures, and infrastructure occurred (Figure 7e). The flooded area was 42.5 ha and 62.8 ha in Garessio and Ceva, respectively; in both towns, very similar dynamics to those observed in 1994, in terms of the type of flood-related processes and damage, were documented. The 2016 flood formation and passage at Clavesana were faster than during the 1994 event. An area of 150.3 ha was flooded, and a maximum of 1.8 m of water depth on the ground level was observed (Figures 7f and 10c). The abstraction weir of the former cotton mill played again a relevant role of interference with respect to the flood propagation; however, no relevant morphogenetic processes involved Gerino. Most out-channel sediment mobilization occurred on the convex side of meanders close to the bank, whereas bank erosion was observed on the concave side, resulting in a slight increase in riverbed width.

*4.4. Building Inundation Analysis*

In this section, a detailed and quantitative assessment of buildings that were inundated during the 1994 and 2016 floods is provided. AFBvf ranged between 33% (Garessio) and 76% (Ceva) in 1994 and between 12% (Garessio) and 29% (Clavesana) in 2016; the lowest values refer to the most recent flood in all three cities (Figure 11).

The total surface of 8.3 ha and 3.5 ha of buildings in Garessio were involved in 1994 and 2016 events, respectively. Figure 12a–c highlights that most of these areas refer to buildings that were built in the period 1933–1994. A progressive increase in CAFB was observed considering the consecutive construction periods (Figure 12a). Referring to the 1994–2016 interval (Figure 12a), the values related to the 1994 flood represent a hypothetical scenario, namely, the extent of the most recent buildings that would have been flooded during the 1994 event. Most of AFBcp refers to 1933–1994 for both the 1994 (Figure 12b) and 2016 events (Figure 12c). Considering the 2016 flood, 3% of AFBcp is associated with the latest time interval, which corresponds to 2.1% of AFBvf_cp. Almost 70% of the AFBcp referred to the 2016 event was built between 1933 and 2016 (Figure 12c). The AFBvf_cp ranged from 29.4 to 40.4% and 2.1 to 16.4% in 1994 and 2016, respectively. It is noteworthy that 50.2% of the total surface of buildings realized over the valley floor during the period 1994–2016 is included within the area inundated by the 1994 flood (Figure 12a).

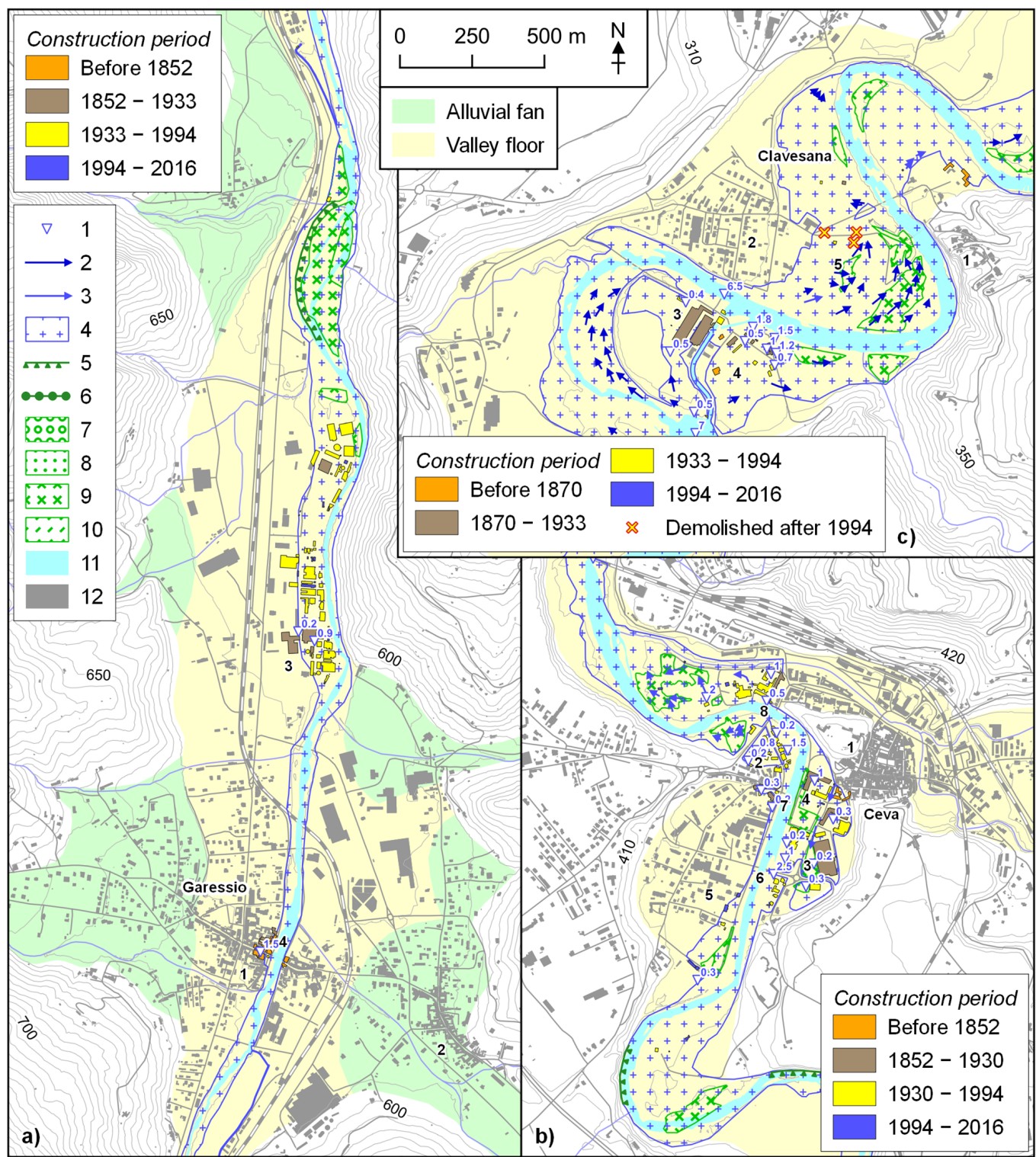

**Figure 10.** The 2016 flood. Flood-related elements and construction period of inundated buildings. (**a**) Garessio. (**b**) Ceva. (**c**) Clavesana. Refer to Figures 5 and 8 captions for the meaning of numbers and symbols, respectively.

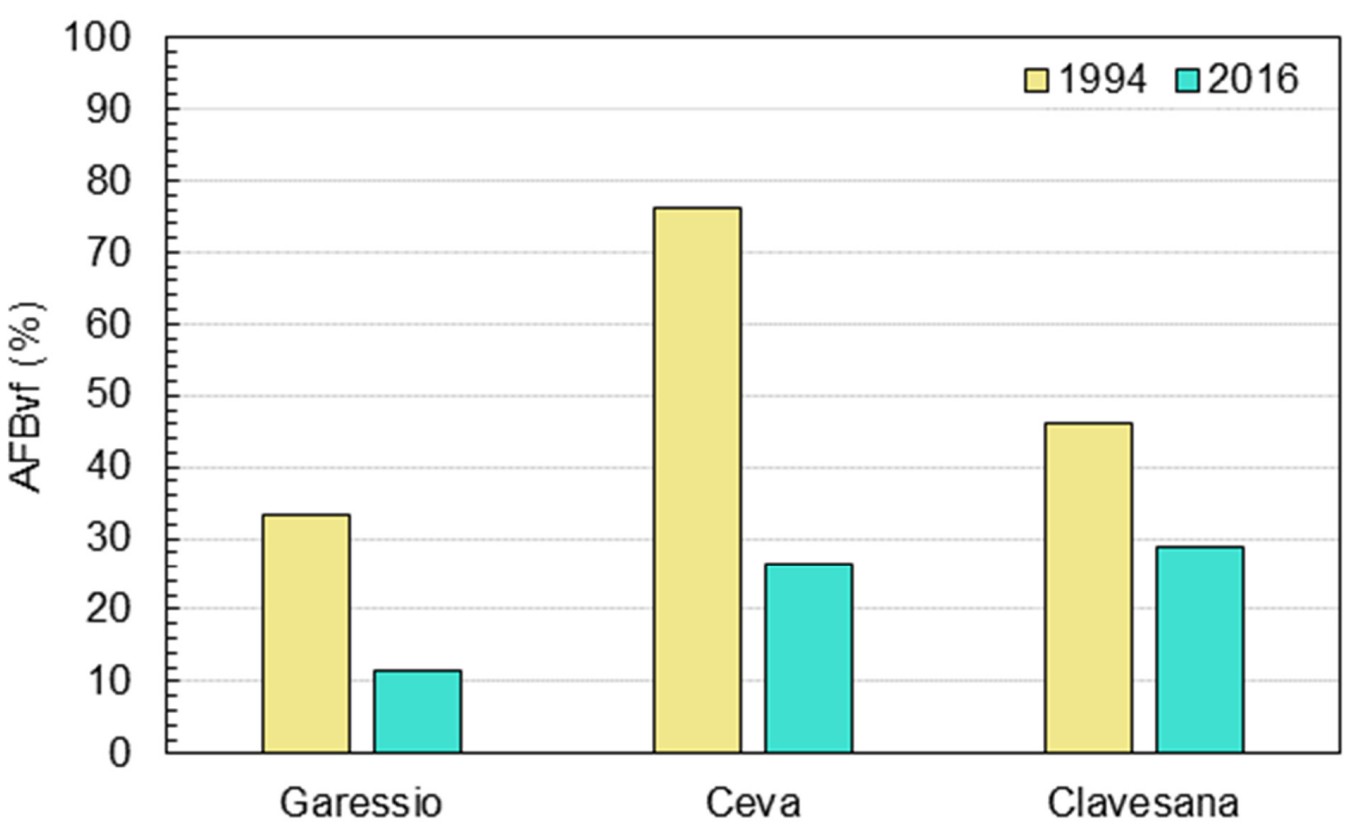

**Figure 11.** Area of flooded buildings with respect to the total area of buildings located within the valley floor (AFBvf) during the 1994 and 2016 floods.

In Figure 12d–f, the results for Ceva are displayed. The total AFB was 8.6 ha and 3.3 ha in 1994 and 2016, respectively. Most of AFB refers to structures dated back to the period 1930–1994 in both the 1994 and 2016 floods (Figure 12e,f), as observed in Garessio, and AFBcp assumed values with reference to the 1994 event that are like those registered in Garessio again (Figure 12b,e). A small surface of buildings dating back before 1852 was flooded in 2016 (Figure 12f). The AFBcp value of 4.5% (Figure 12f) indicates that 0.1 ha of buildings realized after 1994 were affected by the 2016 flood. Moreover, almost 0.7 ha dating back to the latest construction period would have been flooded in 1994 (Figure 12d). Focusing on the AFBvf_cp referred to 1994, the highest value of 100% was noticed for the period before 1852, and values around 70% were observed for the other time spans; AFBvf_cp ranging between 8.9% and 41.6% were observed for 2016 (Figure 12d).

In Clavesana, different trends were documented, as the highest values of AFB and AFBcp were associated with the 1870–1933 construction period for both the 1994 and 2016 floods (Figure 12g–i). Moreover, lower values of AFB than those observed in Garessio and Ceva were overall documented. The total AFB was 2.3 ha and 1.5 ha in 1994 and 2016, respectively; the former value is almost four times lower than the record referred to for the other two cities. No AFB was registered with respect to the most recent construction period for the 2016 flood and the 1994 hypothetical scenario (Figure 12g). After the 1994 event, three buildings located in La Generala quarter and severely damaged by the flood were demolished and relocated (Figures 8c, 10c and 13). This resulted in about 0.2 ha of buildings (620 m$^2$ built in the period 1870–1933 and 1652 m$^2$ in 1933–1994) that were not flooded in 2016. AFBvf_cp ranged from 23.8% to 84.8% and from 0% to 72.4% for the 1994 and 2016 floods, respectively.

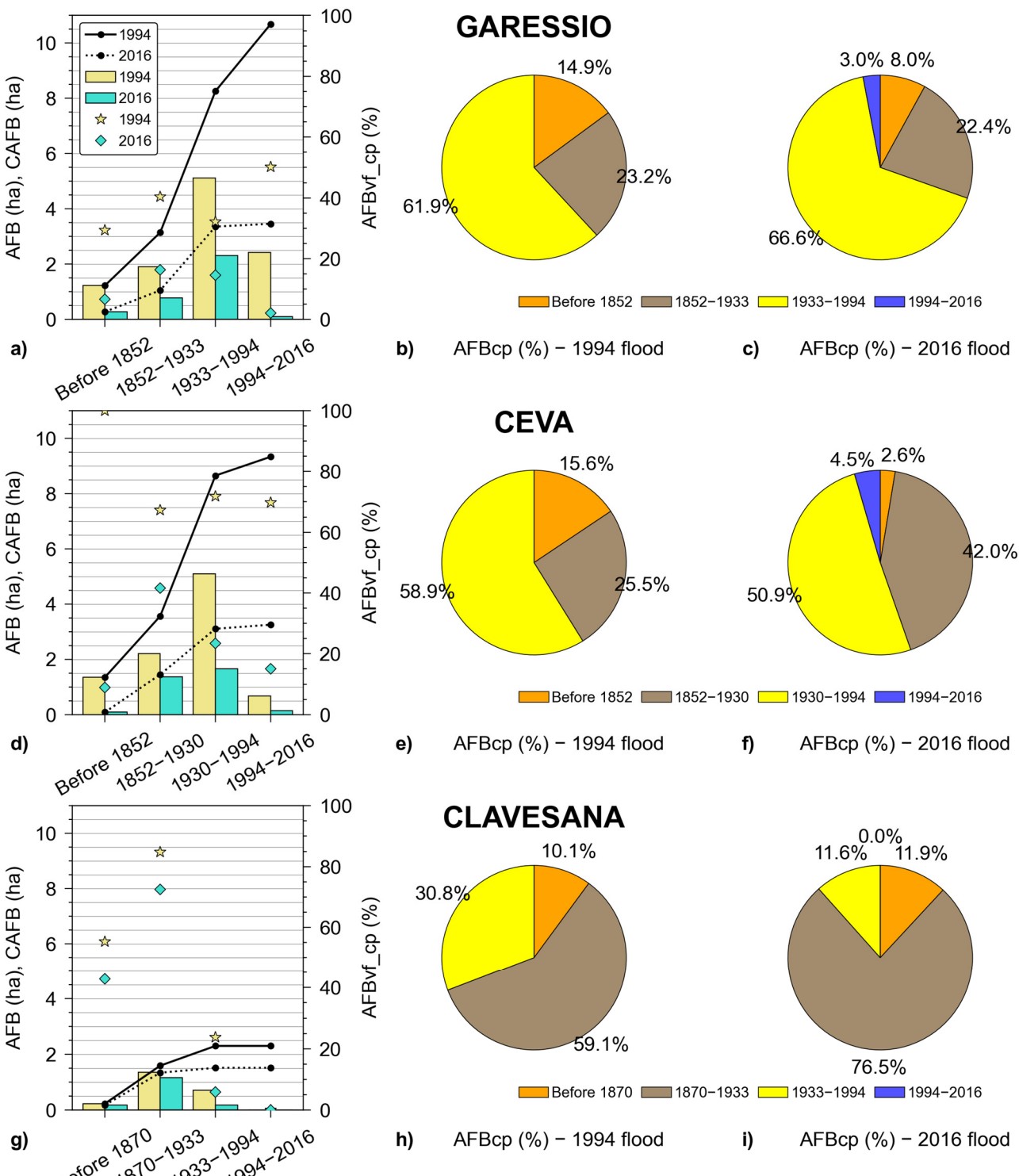

**Figure 12.** Quantitative analysis of buildings that were inundated during the 1994 and 2016 floods for Garessio (**a–c**), Ceva (**d–f**), and Clavesana (**g–i**). AFB: Area of Flooded Buildings; CAFB: Cumulative AFB; AFBvf_cp: Percentage of AFB out of the total surface of buildings located within the valley floor with reference to each construction period; AFBcp: Percentage of AFB referred to construction periods. Bars (indicating AFB) and black points on black lines (indicating CAFB) are keyed to the left *y*-axis; points (indicating AFBvf_cp) are keyed to the right *y*-axis (**a,d,g**). Legend in (**a**) refers also to (**d,g**).

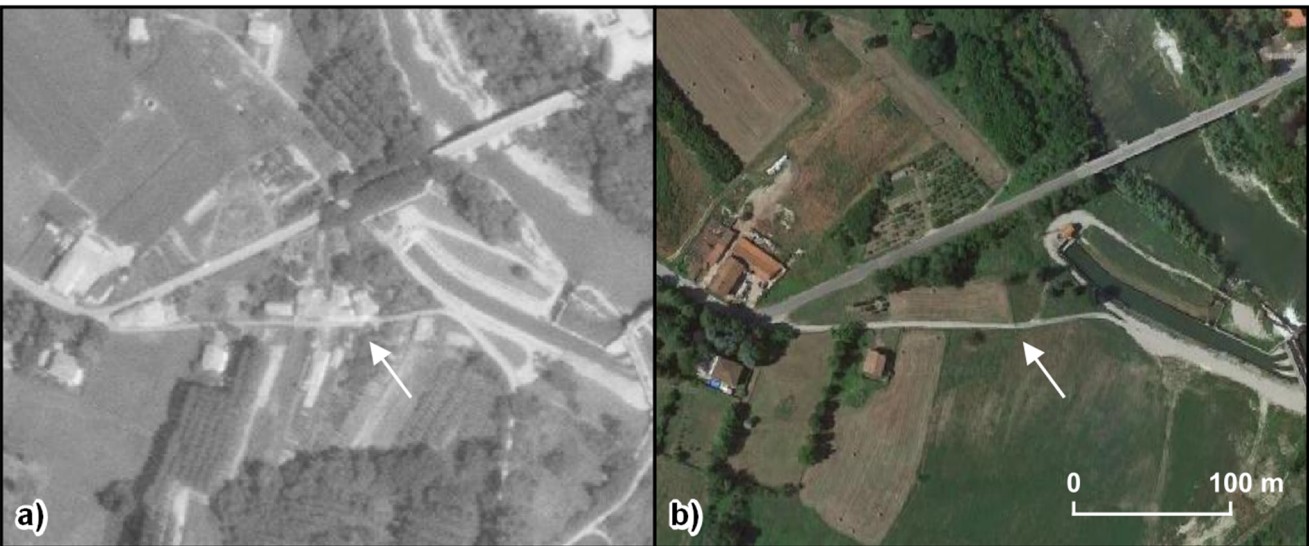

**Figure 13.** The delocalization site in Clavesana (La Generala quarter), indicated by the white arrow, in (**a**) 1988 (orthophoto) and (**b**) today (Google Earth image). Here, exposure was reduced by resettling away from high-risk areas.

In general, major values of AFD were associated with the flood that occurred in 1994 (Figure 12a,d,g). Pie charts in Figure 12 highlight that AFBcp values referring to the most ancient and the most recent periods are never higher than 15.6% and 4.5%, respectively. Considering the urban setting in 2016, an overall building surface of 10.7 ha in Garessio, 9.3 ha in Ceva, and 2.3 ha in Clavesana was located within the 1994 flooded area, while an overall building surface of 3.5 ha in Garessio, 3.3 ha in Ceva, and 1.5 ha in Clavesana was located within the 2016 flooded area. Referring to these values, 22.7% and 3.0% in Garessio, 7.3% and 4.5% in Ceva, and 0% and 0% in Clavesana, for 1994 and 2016 floods, respectively, correspond to surface built-up in the period 1994–2016.

## 5. Discussion

### 5.1. Urban Expansion and Floods

Garessio, Ceva, and Clavesana are characterized by different geomorphological features that reflect their positions along the main valley of the Tanaro Catchment. The first developed in a mountain area, whereas the others arose downstream over a terraced floodplain and close to hills. However, they similarly present a narrow valley floor crossed by the Tanaro River, which is bordered by steep, high slopes and alluvial fans in Garessio and steep, high fluvial scarps delimiting fluvial terraces and slopes in Ceva and Clavesana. These physical settings conditioned the urban expansion of cities, which was rather scarce and, in Ceva and Clavesana only, limited to the fluvial terrace much higher than the valley floor up to the mid-19th century.

During the investigated period, Garessio, Ceva, and Clavesana overall experienced an increase in built-up surface; urbanization took place over the flat and low-gradient areas, namely, the valley floor and alluvial fans in Garessio and both the valley floor and fluvial terraces in Ceva and Clavesana. Urban expansion was seriously fostered by two main events: (i) the realization of the railway line between Bra and Ceva in 1874 and between Ceva and Ormea in 1893, namely, a strategic infrastructure for communications and commerce in the Tanaro Valley; and (ii) the Italian economic boom, namely, the prosperous period of technologic and economic development following the Second World War that resulted in a generalized growth of cities at the national scale during the 1960s and 1970s [118,119]. In these decades, the highway connecting Ceva with the Ligurian seaside, to the south, and Torino, to the north, was realized, further promoting urban expansion.

In all three cities, the first factories settled between the late 19th century and the early 20th century; simultaneously, streets, bridges, and residential buildings were built. However, most buildings were built between the 1930s and 1994, also considering urban expansion over the valley floor only (Table 2). Despite the common temporal evolution of built-up area in the three cities considered, marked differences were documented in the extent of urbanization, with large surfaces overall involved in this process in Garessio and Ceva and much smaller areas in Clavesana. These outcomes are reflected in the anthropogenic conditioning of the Tanaro riverbed. In fact, it is (i) completely channelized, straightened, and encompassed within the urban fabric in Garessio and Ceva, and (ii) channelized by discontinuous bank protections and two abstraction weirs, with a sharply lower presence of buildings close to banks in Clavesana.

The flood-proneness of the valley floor is clearly highlighted by the long history of damaging fluvial floods documented over the last centuries (Table 1). The 5–6 November 1994 and 24–25 November 2016 floods were considered representative because (i) the former was the largest event to occur over the last centuries in these urban centers and the first very high magnitude flood that involved the whole Tanaro valley after the urban expansion of the 19th and 20th centuries, and (ii) the latter was the first high-magnitude event to occur after 1994.

The former event flooded and damaged thousands of buildings and many structures and infrastructures [94]. About 60% of AFBcp refers to buildings realized between the 1930s and 1994 in Garessio and Ceva, namely, the period of the largest building expansion, while most of AFBcp in Clavesana (59.1%) refers to buildings realized in the period 1870–1933, when industrial plants settled over the valley floor (Figure 12).

As documented elsewhere [73,91,120,121], the spatial distribution of flood-induced ground effects and the main flood-flow directions over the valley floor highlighted the interference of anthropogenic linear structures (e.g., roads, railways, walls, and levees) on flood propagation and, thus, on triggering erosional and depositional processes in all three cities considered. According to witnesses, many of these works served as barriers; they locally raised the water level, causing temporary but noteworthy differences in building inundation, and often collapsed, triggering sudden localized flood waves [94]. Abstraction weirs conditioned flood water propagation, fostering flooding and erosional processes over the valley floor in Clavesana (Figures 8c and 9); moreover, bridges represented obstacles for flood water due to channel narrowing associated with bridge piers and large woody debris accumulation upstream [109,114,116].

After the 1994 flood, some interventions were performed for flood hazard reduction, particularly focusing on bridge pier removal and levee construction. Conversely, these contribute to making flood water propagation downstream faster. In the meantime, urban expansion associated with residential and industrial structures continued. In the period 1994–2016, the highest rate of building area increase was observed for all three cities, and 60% in Garessio, 9% in Ceva, and 14% in Clavesana of the area occupied by new buildings was over the valley floor (Table 2). It is noteworthy that buildings built between 1994 and 2016 on the valley floor would have been inundated by 50.2% in Garessio and 69.7% in Ceva by the 1994 flood. No new buildings were realized in the 1994 flooded area in Clavesana.

In 2016, the rainfall causing the flood was different in terms of duration, intensity, and spatial distribution; moreover, the valley floor presented different features influencing flood propagation due to the aforementioned interventions. However, this flood caused severe damage to Garessio, Ceva, and Clavesana, although a lower building surface was inundated (Figure 10). Most of the flooded buildings were built between the 1930s and 1994 in Garessio e Ceva and in the previous period in Clavesana, such as for the 1994 flood. However, the 2016 flood also hit buildings built after 1994 in Garessio (0.1 ha) and Ceva (0.1 ha). The 1994 and 2016 floods hit residential, commercial, and industrial masonry buildings in all three cities investigated. In 2020, the most recent high-magnitude flood hit the Tanaro valley floor again, causing serious damage to Garessio, Ceva, and Clavesana in

a quite similar scenario to that of the 2016 event [109]. After this flood, the Odasso Bridge in Garessio was demolished to remove its barrier effect in the city center.

As reported in Figures 8c and 10c, in Clavesana, three buildings inundated by 3.4 m and seriously damaged in 1994 were completely removed and relocated (Figure 13). Most probably, the high level of damage was decisive for preferring relocation. Moreover, the nursery school in Gerino quarter, rebuilt after the 1994 and 2016 events and further flooded and damaged in 2020, was abandoned due to the recurring events. Relocation of exposed elements is generally uncommon and not encouraged in Italy since there is no adequate financial assistance in this regard and it may incur strong opposition [122–124]. However, this measure may largely contribute to risk reduction by focusing on exposure. Severe damage caused by floods may represent, as in this rare Italian example, the occasion to promote the abandonment of hazardous areas without following the common propaganda slogans for reconstruction "as it was before the flood".

The progressive urbanization of valley-floor areas resulted in an increase in exposure to floods and denoted that historical information was underrated or completely neglected. The historical series of damaging floods highlighted the long-term flood-proneness of the Tanaro River valley floor. The 1994 and 2016 floods clearly demonstrated that urban expansion largely affected floodable areas and that flood-prone areas and fluvial morphodynamics were not properly considered in urban development planning. It is noteworthy that urbanization in flood-prone areas occurred even after the entry into force of regulation about land-use in floodable areas dated back to the mid-1990s and derived from a novel frame of catchment-scale land planning that originated just as a response to the 1994 flood [20,125].

Something similar was observed in Alba, a city close to the Tanaro River downstream of Clavesana, where the surface seriously flooded during four floods in the two-year period 1948–1949 experienced urbanization in the second half of the 20th century [20]. As a result, the 1994 flood of the Tanaro River caused widespread and very severe damage [20]. A recent study focused on flash floods highlighted that the coastal cities of Genova, Olbia, and Livorno (Italy) have a long history of floods and that the degree of flood-caused damage has increased year after year in proportion to the degree of urbanization [46]. These cities experienced severe landscape changes related to the urbanization process over the last 150 years [46]. Roy et al. [123] registered that in the Chaudière River Basin (Québec), new buildings were erected in the floodable zone with a return period of 20 years from 1979 to 1997, partly according to regulations. Furthermore, they observed that buildings located in floodplains were restored and renovated more often than those outside of risk areas as a result of flood damage and thanks to government compensation, which resulted in a greater degree of damage at the next flood. In Wuhan (China), there was a significant increase in flood exposure during the period 1954–2000 because of unplanned and unregulated urbanization, whereas a substantial and continuous reduction in flood exposure between 2000 and 2020 associated with appropriate and integrated land-use planning was documented [80].

*5.2. Recommendations for Urban Expansion*

Generally, urban expansion in flood-prone areas is fostered by institutional bodies and citizens during long periods of undisturbed coexistence of settlements and rivers, namely, in the absence of relevant flood events. This condition makes people lose their local historical memory and, thus, their social perception of flood hazard [20,126–128]. Furthermore, urban expansion is often carried out along with the realization of flood protection structures aiming to reduce flood hazard (Figure 14). In Garessio, Ceva, and Clavesana, levees were built along the Tanaro River to contain flood water, but floods that occurred in 1994, 2016, and 2020 clearly highlighted the vulnerability of these structures. As mentioned in previous research [31,127,129–131], due to structural measures such as flood detention dams and levees, more and more people and economic assets have concentrated in floodplains worldwide, with the assumption that the floodplains are free from flood risks

because of the flood protection scheme. These structures fulfill society's requirement for safety and floodplain development, thinking that these protection projects can completely resist floods [31]. However, paradoxically, they might increase flood risk as they result in major exposure associated with urbanizing protected areas and inducing property owners to invest more in their property, which leads to higher potential damages in case of defense system failure [127,132,133] (Figure 14).

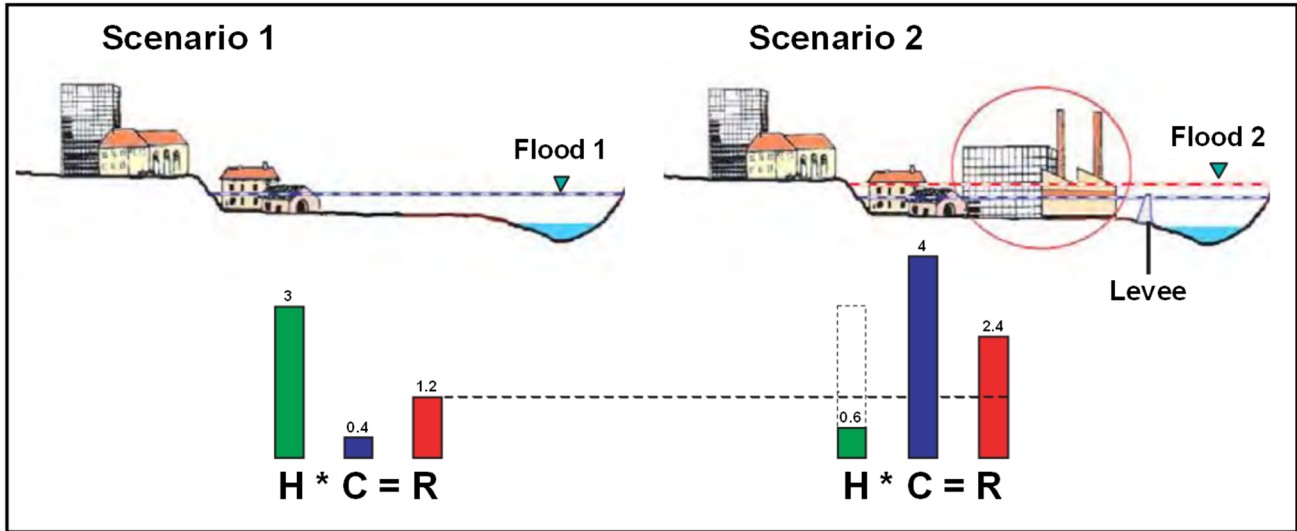

**Figure 14.** Simplified scheme of the "levee-effect". Risk is expressed as the product of hazard (H) and consequences (C), intended as the expected damage derived from the combination of exposure and vulnerability. In scenario 1, the floodplain may be flooded, but damage is relatively low as a low exposure is observed. In Scenario 2, the levee protects from flood 1; however, a higher-magnitude flood, flood 2, may occur, causing the overtopping of the embankment. Furthermore, floods may cause the levee collapse even without overtopping. The misleading perception of safety induced by the flood-defence structure leads to urban expansion over the floodplain (red circle), which results in increased exposure and overall risk. The landscape transformation from scenario 1 to scenario 2 reflects the urban expansion process associated with flood-defence construction that occurred in Ceva and Clavesana from the second half of the 19th century onwards and overall between the 1930s and 1994 (from [132], modified).

Urbanization in flood-prone areas is a critical issue worldwide, and flood management is needed [134,135]. Past floods, their damage, the effects of flood defense based only on structures, and the uncertainty associated with climate change should raise some issues with flood management strategies. Flood control works, unplanned post-flood interventions, and post-disaster assistance, albeit still important, should no longer be the main pillars supporting flood policy, and warning and emergency actions, land use planning, and insurance should acquire a more prominent role [136]. One of the major factors contributing to the rise in urban flood damages is simply the increasing number of people and assets that are physically exposed to floods in cities [80,137,138]. Thus, land use planning for flood control based on historical data on floods and urbanization should be largely implemented, as it allows to drive urban expansion away from potentially floodable areas, preventing exposure [133,139]. Furthermore, flood insurance and compensation systems are important parts of strategies for dealing with flood risk [140–142]. However, in Italy, there is still no law imposing insurance against geo-hydrological processes; in contrast, in some European countries, for example Belgium, Denmark, Switzerland, France, and Spain (even since the 1940s), and non-European countries, such as Japan or the United States, catastrophe insurance has been implemented for several years with results that appear to be satisfactory [143–147].

Today, it is essential to avoid further exposure of vulnerable elements associated with either urban expansion or existing-building renovation in floodable areas of Garessio, Ceva, and Clavesana. Moreover, interventions aiming to remove exposed elements, particularly in the case of key services such as hospitals, police stations, and schools, or to reduce their vulnerability, should be planned and realized from a risk reduction perspective.

*5.3. Integrated Approach and Metrics to Investigate Urbanization and Floods*

In this study, an integrated approach that combines historical information from various sources, earth observation through remote sensing data, and field evidence was implemented. This approach allowed us to investigate past fluvial floods, thus flood-proneness, urbanization processes, flood-induced ground effects, flood dynamics [91], and building exposure to floods.

This work highlighted the great importance of freely available historical maps, high-resolution freely available aerial and satellite images, and free and open-source GIS software to perform repeatable, low-cost, highly accurate, and multitemporal analyses concerning both flood events and urban evolution. Moreover, (i) extensive post-flood field surveys to depict in detail the flood scenario and validate post-flood remote-sensing data, and (ii) field surveys to verify the present-day urban setting and identify the main geomorphological features of the Tanaro River valley floor were essential. Against the uncertainty affecting the frequency and magnitude of floods in the general context of climate change and anthropogenic landscapes [148–151], the analysis of trusted data related to past floods constitutes a relevant source of reliable information. The overall method implemented in this study is not limited to specific geographical areas and does not require large facilities or specific equipment. Data availability is the main limitation. The phases of archival research on flood data and the definition of building vector layers were rather time-consuming and error-inducing. Furthermore, the combination of maps and imagery presenting various scales may introduce uncertainties during the interpretation phase. For this reason, highly accurate georeferencing procedures are recommended.

A set of GIS-based metrics considering the building footprint were introduced to investigate (i) the urbanization process and (ii) the extent and construction period of inundated buildings. These metrics enlarged the spectrum of available tools for such analyses and allowed for detailed multitemporal investigations at the spatial scale of the individual building. This spatial scale provides accurate and essential data on building location, extent, and construction period for further analyses concerning vulnerability and risk assessment.

## 6. Conclusions

Garessio, Ceva, and Clavesana were hit by damaging floods several times over the last centuries, demonstrating the flood-proneness of the valley floor. These cities overall experienced an increase in built-up surface from the second half of the 19th century onward, especially between the 1930s and 1994. The physical setting of the Tanaro River valley, the construction of transport infrastructure, and the Italian economic boom were identified as driving factors for urban expansion. Conversely, floods did not condition the occupation of flood-prone areas. Urbanization affected the valley floor and was often associated with the construction of flood defenses and riverbed channelization. As a result, a relevant increase in exposure to floods over time, related to the building occupation of the valley floor, was documented against a general carelessness of historical information on floods. The 1994 and 2016 high-magnitude floods severely hit large areas with residential, commercial, and industrial masonry buildings. These events highlighted (i) that anthropogenic structures conditioned flood propagation and triggered erosional and depositional processes, (ii) that urban expansion clearly affected floodable areas, and (iii) that flood hazard was not properly considered in urban development planning, even in recent times. However, relocation of some elements exposed to floods was implemented in Clavesana after the 1994 and 2016 floods for risk reduction. This was one of the rare cases of relocation in Italy. As

for future urban development, further exposure of vulnerable elements in floodable areas of Garessio, Ceva, and Clavesana should be avoided. Moreover, interventions aiming to reduce exposure or vulnerability should be planned and realized from a risk reduction perspective.

This research reveals the relevance of information derived from multiple sources on past floods and urbanization processes from a land planning and land management perspective. Furthermore, the need for land use planning for flood control aiming to drive urban expansion away from potentially floodable areas is pointed out.

An integrated approach for investigating urbanization, flood-proneness, and exposure to floods was developed. Collection and review of historical data, photograph interpretation, GIS analysis, and field surveys were performed and combined. Furthermore, some GIS-based metrics focusing on building footprint were introduced to quantitatively assess urban expansion and building exposure to floods at the individual-building scale.

The outcomes from this study represent an essential knowledge base for technicians and policymakers to plan urban development and inform effective and sustainable management measures aiming to mitigate hydrogeomorphic risk. Dissemination of the results is essential to increasing citizens' knowledge and awareness of urbanization and flood issues. A large-scale replication of this research in urban areas would enlarge the spectrum of available information for risk assessment.

This study constitutes the first combined analysis of floods and urbanization. Further work to develop this research should be devoted to (i) investigating the Tanaro River morphology and dynamics with reference to catchment-scale hydrogeomorphic processes, (ii) characterizing in detail the elements exposed to floods for vulnerability assessment, (iii) implementing flood-simulation models for hazard analysis, and (iv) assessing risk conditions considering various flood scenarios.

**Author Contributions:** Conceptualization, F.F., F.L., A.M. and L.T.; data curation, B.B. and A.M.; funding acquisition, F.F. and L.T.; investigation, A.M.; methodology, F.L., A.M. and L.T.; project administration, F.F. and L.T.; supervision, F.F., F.L. and L.T.; validation, F.F. and A.M.; visualization, F.L. and L.T.; writing—original draft preparation, F.F., F.L., A.M. and L.T.; writing—review and editing, B.B., F.F., F.L., A.M. and L.T. All authors have read and agreed to the published version of the manuscript.

**Funding:** This research was funded by (i) the FONTES PRIN Project (Italian National Research Project "Fonti geostoriche e sistemi informativi per la conoscenza del territorio e la gestione dei rischi ambientali e culturali") (https://fontes.univr.it/ (accessed on 28 July 2023)), supported by government funding (Project CNR Number DTA.AD003.737), and (ii) the DM 1062/2021 FSE REACT-EU PON Project—Green (CUP N. D31B21008270007 University of Genova).

**Data Availability Statement:** Data sharing is not applicable to this article.

**Acknowledgments:** The authors are grateful to C. Giampani (ARPA Piemonte) for sharing materials and to A. Acquarone, G. Galliano, and G. Susella for photo courtesy. They also thank the reviewers for their constructive comments and suggestions.

**Conflicts of Interest:** The authors declare no conflict of interest.

## Abbreviations

The following abbreviations are used in this manuscript:

| | |
|---|---|
| AFB | Area of flooded buildings |
| AFBcp | Areas of flooded buildings refer to construction periods |
| AFBvf | Area of flooded buildings out of the total area of buildings located within the valley floor |
| AFBvf_cp | Area of flooded buildings out of the total area of buildings located within the valley floor with reference to construction periods |
| BA | Building area |

| BAvf | Building area over the valley floor |
| BAvf Ratio | Building area over the valley floor ratio |
| BE Rate | Rate of building expansion |
| BEvf Rate | Rate of building expansion over the valley floor |
| CAFB | Cumulative area of flooded buildings |
| CBA | Cumulative building area |
| CBAvf | Cumulative building area over the valley floor |

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
