# Peer review of "Integrated Approach for the Study of Urban Expansion and River Floods Aimed at Hydrogeomorphic Risk Reduction"

_remotesensing, doi:10.3390/rs15174158_

Round 1

Reviewer 1 Report (Previous Reviewer 4)

I am glad with the authors' comment to all my questions and now I feel that the paper deserve to be published.

minor English check is required in the Introduction for punctuation.

Author Response

Reply to Reviewer#1

I am glad with the authors' comment to all my questions and now I feel that the paper deserve to be published.

We thank the reviewer for his comment and for his previous suggestions that allowed us to improve the manuscript.

Reviewer 2 Report (Previous Reviewer 2)

The authors have made extensive revisions, but there are still some writing problems.

1. The author mentions that the integrated approach is a synthesis of historical information, remote sensing information, and field data, but the content is a compilation and organization of information, not an integrated analytical approach. The integrated approach should be to analyze the relationship between flooding and urban expansion by considering multiple influencing factors.

2. It is suggested that the authors analyze the relationship between the area of different flood occurrences and the architectural characteristics of the area where the flood occurs.

3. It is suggested that the discussion of what deficiencies and shortcomings still exist in this study, which can provide guiding research directions and ideas for scholars related to this study.

4. The conclusion needs to continue to be revised and lacks a summarized research conclusion. Characteristics of urbanization development? Characteristics of flood occurrence? Characteristics of buildings in flood-prone areas? None of them are clearly reflected in the conclusion.

Minor editing of English language required

Author Response

Reply to Reviewer#2

We thank the reviewer for his comments and suggestions that allowed us to improve the manuscript.

We have revised our manuscript following the reviewer’s suggestions and we have answered his questions.

The response to the reviewer’s comments, along with the explanations and justifications on the changes incorporated in the manuscript are listed below (all the changes and additions can be viewed within the attached tracked version of the manuscript).

The authors have made extensive revisions, but there are still some writing problems.

---

1. The author mentions that the integrated approach is a synthesis of historical information, remote sensing information, and field data, but the content is a compilation and organization of information, not an integrated analytical approach. The integrated approach should be to analyze the relationship between flooding and urban expansion by considering multiple influencing factors.

We think that the term “integrated approach” may include different concepts and shades, including the contents of the present paper. This research is developed around the relationship between urban expansion and floods. Various factors are considered to explain urbanization (physical, economic, social) even if the quantitative analysis is strictly focused on buildings and floods. Furthermore, our work is based on multiple data sources and various techniques. Thus, we adopted an approach, a method, which combines (integrates) several types of data and analysis (integrated approach of historical analysis, hydrogeomorphic investigations, and geospatial techniques); this approach analyses the relationship between flooding and urban expansion by considering multiple data and factors.

2. It is suggested that the authors analyze the relationship between the area of different flood occurrences and the architectural characteristics of the area where the flood occurs.

The analysis of individual buildings (in terms of architectural characteristics) goes beyond the scope of this research and, as specified in Conclusions, this could be a future development of our work. However, we indicated that the 1994 and 2016 floods hit residential, commercial, and industrial masonry buildings in all the three cities investigated.

3. It is suggested that the discussion of what deficiencies and shortcomings still exist in this study, which can provide guiding research directions and ideas for scholars related to this study.

This part was added in the first round of revision at the end of Conclusions, according to the reviewers’ comments.

4. The conclusion needs to continue to be revised and lacks a summarized research conclusion. Characteristics of urbanization development? Characteristics of flood occurrence? Characteristics of buildings in flood-prone areas? None of them are clearly reflected in the conclusion.

The first paragraph of Conclusions summarizes the main outcomes of our research. The main elements associated with each issue considered are reported. As for characteristics of urbanization development, we indicated where and when urbanization took place. Moreover, the driving factors for urban development were indicated. We do not properly understand “characteristics of flood occurrence”. We specified that anthropogenic structures conditioned flood propagation and triggered erosional and depositional processes. Moreover, we referred to the historical series of floods. According to the reviewer’s request we have added the characteristics of buildings in flood-prone areas (please, see the second response). Moreover, we have added a specific part focused on recommendations for urban expansion (that is associated with a new part in section 5.2, as requested by reviewers).

Reviewer 3 Report (New Reviewer)

Comments and Suggestions

Major Comments

Line 222: The maps of buildings have different scales, datums and sources (some maps are very old), which were manually digitized at a constant scale of 1:2000.There are possibilities of uncertainties during the processing of maps and further analysis. Please comment on the reliability of produced maps, and justify how you ensured their correctness.

Line 744: The authors are strongly suggested to provide further recommendations and suggestions for local urban expansions of three cases (considering discussed results) to avoid future floods. The results analyzed based on an integrated approach of historical urban expansion maps, flood information, remote sensing data, and field information may provide practically applicable solutions for urban development to reduce future flood hazards.

Minor Comments

Line 39-41: What is the difference between “ anthropogenic interventions” and “man’s use of catchments”? Please check and improve/describe the later part of sentence.

Line 72: The authors are suggested to also mention other key drivers of floods, for example, climate change, etc., and then may can focus on “Urbanization/urban expansion” as one of the main key drivers of flooding, and highlight it as a research gap of study.

Line 267: Please provide reference or clarification about the flood classification, as it was performed by authors or guidelines followed from any published/unpublished document.

Author Response

Reply to Reviewer#3

We thank the reviewer for his comments and suggestions that allowed us to improve the manuscript.

We have revised our manuscript following the reviewer’s suggestions and we have answered his questions.

The response to the reviewer’s comments, along with the explanations and justifications on the changes incorporated in the manuscript are listed below (all the changes and additions can be viewed within the attached tracked version of the manuscript).

Line 222: The maps of buildings have different scales, datums and sources (some maps are very old), which were manually digitized at a constant scale of 1:2000.There are possibilities of uncertainties during the processing of maps and further analysis. Please comment on the reliability of produced maps, and justify how you ensured their correctness.

We have revised following the reviewer’s request. We have introduced a specific part at the beginning of section 3 and in section 5.3.

The constant scale was considered just for visualization, namely, to make better and systematic the digitizing procedure.

Line 744: The authors are strongly suggested to provide further recommendations and suggestions for local urban expansions of three cases (considering discussed results) to avoid future floods. The results analyzed based on an integrated approach of historical urban expansion maps, flood information, remote sensing data, and field information may provide practically applicable solutions for urban development to reduce future flood hazards.

This comment is not very clear. Line 744 refers to the section 5.3, which is specifically focused on the method and the approach. Thus, management issues related to the three cases could be inappropriate in this part.

However, to address the reviewer’s comment we have added some recommendations and suggestions in section 5.2 (which was thought to be more suitable for this purpose) being aware that further analyses are needed to define in detail effective and sustainable solutions for flood risk management.

Line 39-41: What is the difference between “ anthropogenic interventions” and “man’s use of catchments”? Please check and improve/describe the later part of sentence.

We have revised following the reviewer’s request. These terms are something similar, they were considered to say that anthropogenic conditioning was both along riverbeds and over catchments. We have rephrased in order to make it more understandable.

Line 72: The authors are suggested to also mention other key drivers of floods, for example, climate change, etc., and then may can focus on “Urbanization/urban expansion” as one of the main key drivers of flooding, and highlight it as a research gap of study.

We have modified this part to improve the text. However, this part was revised (in the first revision round) just to focus on urbanization and fluvial flood hazard only, according to the reviewer’s request. Aiming to maintain the focus on this issue, we would avoid adding further comments. Here we want to underline that there are two ways to investigate urban expansion and floods: 1) urban expansion as a driver of floods; 2) floods as a driver of urban expansion. This twofold vision is necessary to discuss the research gap and to introduce this work. Climate change as a flood driver would need a detailed and longer discussion that goes beyond (off-topic) the reference frame of this introduction.

Line 267: Please provide reference or clarification about the flood classification, as it was performed by authors or guidelines followed from any published/unpublished document.

We have considered the approach described in [91]. The meaning of “flood classification” is not clear. The considered floods were not classified (we do not understand how/what should be classified). According to water levels and inundated area extension these floods were identified as high-magnitude events. Flood-induced ground effects and flood-water dynamics were classified according to [91]. This is indicated within the text.

Reviewer 4 Report (New Reviewer)

Please find comments in document attached 

Minor grammatical errors 

Author Response

Reply to Reviewer#4

(Please, see the reviewer’s detailed report)

We thank the reviewer for his detailed report and positive feedback.

Reviewer 5 Report (New Reviewer)

Dear Editor.

As I understand form the incorporated revisions and corrections in the text, this is a resubmission. In my opinion,t he paper is very qualitative and do not needs further revisions.  

Author Response

Reply to Reviewer#5

Dear Editor.

As I understand form the incorporated revisions and corrections in the text, this is a resubmission. In my opinion,t he paper is very qualitative and do not needs further revisions.

We thank the reviewer for his comment.

This manuscript is a resubmission of an earlier submission. The following is a list of the peer review reports and author responses from that submission.

Round 1

Reviewer 1 Report

Urbanization and floods are closely related. The topic is very interesting, but there are some academic problems that should be considered.

(1) In section 4.2, I would like to see the temporal and spatial distribution of rainfall and the flood hydrograph.

(2) How was the inundated area obtained during the flood events?

(3) The Discussion section is too long. I cannot see the focus of the discussion.

(4) What is the main contribution of this research?

(5) The conclusion part should be summarized precisely.

(6) The title is ‘integrated approach…’, what does it mean? I cannot see integrated approach throughout the manuscript.

Reviewer 2 Report

Floods can damage urban infrastructure, cause significant property damage and endanger the lives of urban residents, thus affecting the sustainable development of cities. The study of the relationship between flooding and urbanization process is of great significance, as it can provide a good basis for scientific urban planning and land management in flood-prone areas, thus reducing the hydrogeomorphic risks in urban areas.

The paper collects detailed historical and textual data on urban expansion and major floods in three cities in the southern Italian mountains, and provides a comprehensive and detailed description of the urban expansion characteristics, the hydrogeomorphic conditions of floods and the damages caused, as well as an analysis of the relationship between floods and the spatial layout of urban buildings. However, there are still some problems that cannot be ignored in this thesis, which are mainly reflected in the fact that the thesis only provides a rough description of the flooding and urbanization process in the study area with the collected data, but does not conduct an in-depth quantitative study on the evolution of urban development and its spatial layout, as well as the flooding influence mechanism on the evolution of urban spatial layout. In short, there are more qualitative descriptions and fewer quantitative studies; specifically, they are reflected in the following aspects:

1、The application of remote sensing to solve flood simulation and disaster damage assessment is the theme of the special issue, but the paper only analyzes the relationship between flood occurrence and urban development qualitatively, but does not make good use of the spatial information acquisition capability of remote sensing and the spatial analysis function of GIS to conduct quantitative research.

2. In line 83 of the paper, the issue of "few works focusing on the evolution of urban settlements in flood-prone areas, in terms of building presence over time, are available"is raised, but it lacks an explanation of the research progress on the relationship between flood occurrence and urbanization, especially for the mountainous cities in southern Europe, the proposal of this question lacks basis. The thesis should address the shortcomings of the previous research work, and condense the scientific problem to be solved; in view of the serious shortcomings of this thesis, the innovation and necessity of this research is doubtful.

3. Why did the thesis choose the cities of Garessio, Ceva, and Clavesana as the study area? Are they representative? Do they have their own characteristics in terms of topography, urban layout, and flood occurrence? These are not clearly stated in the text, and the paper just mixes them together for elaboration, which is not appropriate.

4、For the response of urbanization expansion to the occurrence of floods, the title of the thesis emphasizes the use of a comprehensive analysis method, which should involve the urban planning design and implementation of flood relief area setting, construction of facilities such as flood control levees, design of buildings to resist flooding, and flood relief channel obstacle removal measures, etc. These aspects are not reflected clearly enough in the paper.

5. The urban expansion data collected in the paper is from 1952-2016, and the flood occurrence data is from 1610-1906, while the paper only focuses on the analysis of flood occurrence and urbanization process in 1994 and 2016, other data do not play their proper role, so why should them be included in the paper?

6、There are still some writing issues with the paper, such as the paper put the effect of flooding occurrence on the layout of urban buildings in the discussion section, they should be put in the study results. The discussion from multiple perspectives mainly focuses on the evolution pattern of urban building layout in the mountainous cities in southern Europe of the study results, the spatial change trend of flooding occurrence in the region, and the mechanism of the impact of flooding occurrence on urban building layout. The conclusion of the study should also cover the above three aspects, but the conclusion of the paper is rather vague.

Moderate editing of English language required

Reviewer 3 Report

The presented study compiles and analyzes information to highlight errors in land use planning for flood control aiming to steer urban expansion away from flood-prone areas.  It emphasizes the importance of conducting such an analysis before urban planning but does not introduce new techniques or innovative components.

The abstract mentions novel metrics for quantitative analysis of urbanization and flooding, but this novelty is not justified within the text.

Table I lists the geographic information used, ranging from scales of 1:10,000 to 1:25,000 and in the text they writed that "The buildings were recovered as polygons from regional vector maps (Table 1) and manually digitized at a constant scale of 1:2000 by a single operator". It's a confusing wording.

The software used is not mentioned.

GIS analyzes are not clearly justified.

Field surveys were conducted to validate GIS data are not shown.

The GIS applications used are not justified as innovative.

Reviewer 4 Report

The authors adopt an integrated approach that combines historical data analysis, field surveys, and Geographic Information System (GIS) investigations. This multidisciplinary approach enables a thorough examination of the flood-proneness, urbanization process, extent and construction period of inundated buildings, as well as the ground effects and dynamics of two representative floods in 1994 and 2016. Overall, this paper represents a good contribution to the field of urban hydrogeomorphic risk reduction.

You need to use shorter sentences in the Introduction for clearer understanding.

Figures 4,6,11,13,14 should be increased for easier reading.

Due to the number of variables, authors should consider including a list of abbreviations.

Provide a unique section describing the Methods components and maybe all the definitions.

Provide a table or a figure about the components to clarify their relationships in this study.

Can you state the objective of the work clearly?

I am concerned about the model validation results. Have you verified them?

Can you explain your conclusions in plain language where general people understanding air pollution can understand these points?

When you talk about climate change and its impact on line 38-39 I suggest to cite some more recent papers in the field:

Wang, K.; Zhou, Y.; Han, J.; Chen, C.; Li, T. Long-Term Tibetan Alpine Vegetation Responses to Elevation-Dependent Changes in Temperature and Precipitation in an Altered Regional Climate: A Case Study for the Three Rivers Headwaters Region, China. Remote Sens. 2023, 15, 496. https://doi.org/10.3390/rs15020496

Todorov, V.; Dimov, I. Unveiling the Power of Stochastic Methods: Advancements in Air Pollution Sensitivity Analysis of the Digital Twin. Atmosphere 2023, 14, 1078. https://doi.org/10.3390/atmos14071078

Wang, L.; Bai, C.; Ming, J. Current Status and Variation since 1964 of the Glaciers around the Ebi Lake Basin in the Warming Climate. Remote Sens. 2021, 13, 497. https://doi.org/10.3390/rs13030497

 Dimov, I. et al. (2022). Optimized Quasi-Monte Carlo Methods Based on Van der Corput Sequence for Sensitivity Analysis in Air Pollution Modelling. In: Fidanova, S. (eds) Recent Advances in Computational Optimization. WCO 2020. Studies in Computational Intelligence, vol 986. Springer, Cham. https://doi.org/10.1007/978-3-030-82397-9_20

Several important questions about your paper arises.

In your study, you examined the extent and construction period of inundated buildings during the 1994 and 2016 floods. Could you provide some insights into the factors that influenced the expansion of urban areas in flood-prone regions? Were there any notable trends or patterns observed?

The paper mentions a rare case of relocation of elements exposed to floods. Could you provide more details about this case? What were the factors that led to the decision to relocate, and what were the outcomes in terms of reducing flood risks?

The study emphasizes the relevance of historical flood information and urbanization data for land planning and management. How do you envision this information being integrated into practical land use planning processes? Are there any specific recommendations or guidelines you would propose based on your findings?

Considering the increasing risks associated with climate change, do you believe that your integrated approach and the strategies developed in this study will be adaptable and effective in the future? Are there any additional measures that you would recommend for long-term hydrogeomorphic risk reduction?

Were there any unexpected or surprising findings during your research that challenged existing assumptions or theories related to urban expansion and flood risk? How did these findings contribute to advancing our understanding of the topic?

Could you provide some insights into the challenges and limitations encountered during the data collection, analysis, and modeling processes? Were there any specific data gaps or uncertainties that affected the robustness of your findings? How did you address these challenges?

In terms of policy implications, what are the key takeaways from your study that can guide decision-makers and land managers in effectively addressing hydrogeomorphic risks in flood-prone areas? Are there any specific policy changes or interventions that you would recommend based on your research?

Looking ahead, are there any plans for further research or follow-up studies based on the findings presented in this paper? Are there any specific aspects or regions that you believe warrant additional investigation to enhance our understanding of urban expansion and flood risk reduction?

Include in the Conclusion the answers of the last questions.

English extensive editing is necessary, on some places it is difficult to understand, especially in the Introduction, use shorter sentences.